# Modeling *Toxoplasma gondii*-gut early interactions using a human microphysiological system

Carlos J. Ramírez-Flores[1,2], Nicole D. Hryckowian[1], Andrew N. Gale[1], Kehinde Adebayo Babatunde[3], Marcos Lares[3], David J. Beebe[3,4,5], Sheena C. Kerr[4,5], Laura J. Knoll [1]*

1 Department of Medical Microbiology and Immunology, University of Wisconsin-Madison, Madison, Wisconsin, United States of America, 2 Department of Infectomics and Molecular Pathogenesis, Center for Research and Advanced Studies of the National Polytechnic Institute (Cinvestav), Gustavo A. Madero, Mexico City, Mexico, 3 Department of Pathology and Laboratory Medicine, University of Wisconsin-Madison, Madison, Wisconsin, United States of America, 4 Department of Biomedical Engineering, University of Wisconsin-Madison, Madison, Wisconsin, United States of America, 5 Carbone Cancer Center, University of Wisconsin-Madison, Madison, Wisconsin, United States of America

* ljknoll@wisc.edu

## Abstract

Oral transmission of parasites via environmentally resistant cyst stages in contaminated food or water is a common route of human infection, but there are no effective vaccines available for any enteric parasitic infection. Our knowledge of parasite cyst stage conversion and interaction with the intestinal tract is limited. Here, we investigate infection dynamics of *Toxoplasma gondii* cyst-stage in murine jejunum and human intestinal microphysiological systems. We focus on parasite ingress, replication, and conversion of the cyst stage to the rapidly replicating dissemination stage. *In vivo* bioluminescent imaging of mice fed cysts revealed spots of infection throughout the jejunum and ileum, which were selected for further analyses. Immunostaining showed parasite migration and replication predominantly in the stroma, with minimal replication in enterocytes. We recapitulated bradyzoite infection in human intestinal microphysiological systems and showed stage conversion and migration through collagen. This integrated approach elucidates complex host-parasite interactions, highlighting the value of microphysiological systems in advancing understanding and identifying potential therapeutics.

## Author summary

*Toxoplasma gondii* is a widely distributed parasite that infects many people around the world. Its infection is linked to behavioral and metabolic disorders and can lead to serious health complications, particularly in individuals with weakened immune systems. In this study, we investigate how the parasite infects the intestine in the early days of infection and begins to spread within the gut in mice. We focus on the dormant form of the parasite, known as bradyzoites, and examine when they transform into the actively replicating tachyzoite stage within mouse intestines. Our *in vivo* data reveal that *T. gondii*

**Data availability statement:** All data are available in the main text or the supplementary materials.

**Funding:** This work was supported by a Food Research Institute (LJK), National Institutes of Health National Institute of Allergy and Infectious Diseases 1R01AI172874 (LJK, SCK, and DJB), and a Ruth L. Kirschstein Postdoctoral Individual National Research Service Award from the National Institutes of Health National Institute of Allergy and Infectious Diseases F32 AI172084 (NDH). University of Wisconsin Carbone Cancer Center Support Grant NIH P30CA014520. The funders had no role in study design, data collection and analysis, decision to publish, or preparation of the manuscript.

**Competing interests:** We have read the journal's policy, and the authors of this manuscript have the following competing interests: DJB holds equity in Bellbrook Labs LLC, Tasso Inc., Salus Discovery LLC, Lynx Biosciences Inc., Stacks to the Future LLC, Turba LLC, Flambeau Diagnostics LLC, and Onexio Biosystems LLC. DJB is also a consultant for Abbott Laboratories. The other authors declare that they have no competing interests.

primarily replicates in the intestinal stroma during early infection. To further understand this process, we used microphysiological systems to model intestinal infection in humans. This innovative approach allowed us to track the conversion between parasite stages and their subsequent dissemination. Our findings highlight the complex interactions between the host and the parasite, demonstrating the potential of microphysiological systems to uncover critical aspects of *T. gondii* infection. This research enhances our understanding of how this parasite spreads in humans and may inform future strategies for prevention and treatment.

## Introduction

*Toxoplasma gondii* is a cosmopolitan parasite that infects nucleated cells in warm-blooded animals [1,2]. It is estimated that approximately one-third of the human population is infected by *T. gondii* [3]. In the United States, 20–30% of people are seropositive; in contrast, in Brazil and France, it is estimated that more than half of their population are seropositive [3–7]. This widespread distribution has led many to consider *T. gondii* to be one of the most successful parasites known in nature [8]. Despite high rates of infection, there are currently no vaccines available for humans, and treatment options are limited. A live attenuated vaccine called ToxoVax (MSD, New Zealand) effectively reduces losses in breeding sheep, highlighting the potential for future human vaccines [9].

Human infection typically occurs through ingestion of undercooked meat containing tissue cysts, or through water or food contaminated with mature oocysts released in feces from felines, such as the domestic cat [10–13]. *T. gondii* infection in healthy hosts typically manifests as an asymptomatic infection characterized by parasite encystment in muscular tissue and brain tissue [13–15]. However, in immunocompromised individuals, parasite reactivation can lead to chorioretinitis and blindness, or even fulminant encephalitis [16,17]. In pregnant individuals, *T. gondii* infects the fetus, causing congenital toxoplasmosis with severe outcomes like hydrocephalus, chorioretinitis, and intellectual disabilities [18,19].

Tissue cysts contain bradyzoites, a life stage characterized by the presence of amylopectin granules, and the expression of specific markers including BAG1, LDH2, ENO1, and BRP1 [20–24]. Bradyzoites are known for their slow-replicating nature, contributing to the parasite's ability to persist chronically in infected hosts [15,22]. Upon consumption, cysts undergo digestion by pepsin in the host's stomach [15,25–28]. A current well regarded theory states that upon reaching the small intestine, parasites promptly infect enterocytes, revert to the rapidly replicating tachyzoite stage, disseminate throughout the host, and eventually develop into the cyst stage, which is a hallmark of chronic infection [10]. While this infection progression has been studied in mouse models, the initial interactions in the small intestine following the ingestion of cysts is not yet well understood.

Oocysts contain eight infectious sporozoite parasites, which have been shown to rapidly infect ileal enterocytes, migrate to the lamina propria, and convert to tachyzoites within 24 hours [29,30]. Subsequently, parasites replicate in the lamina propria, spread to various organs, and reach the brain by day 6 post-feeding [29,30]. While intestinal bradyzoite infection appears to share similarities with intestinal sporozoite infection, our understanding of early bradyzoite infection in the small intestine is more limited. In mice fed bradyzoites, most of the early infection occurred in the distal half of the small intestine. The transition from bradyzoites to tachyzoites is suggested to appear 12–18 hours after feeding, as evidenced by parasite morphological changes in micrographs [31]. Within 1–3 days post-feeding, parasites were found replicating in both enterocytes and the lamina propria, with an increased parasite number in the intestine after 2–3 days. Subsequently, parasites were detected in the brain 6–7 days post-feeding [31].

Studying the first 3 days of the bradyzoite infection process in mouse intestines has proven challenging, hindering a comprehensive understanding of this process. Studies have focused primarily on the host immune response within the intestine after 3 days post-infection [32,33]. Gregg *et al* observed infected villi in the proximal jejunum of mice fed cysts 3 days post-consumption with the presence of macrophages, neutrophils, and monocytes in the lamina propria [32]. Coombes *et al* described the infection in the distal third of the small intestine of mice fed cysts [33]. By 5–6 days post-feeding, single parasites were detected within villi, with scarce locations containing already replicating parasites, often concentrated at the villi tips [33]. Despite the various approaches aimed at understanding early bradyzoite infection in the mouse model, our comprehension of processes such as bradyzoite release, interaction with the intestine in the early phases of the infection, parasite motility through mucus, villi infection, parasite conversion, and dissemination through the intestine and the organism remains limited. This limitation is particularly pronounced in humans, where the infection dynamics and interactions have obviously been limited to cell lines. Addressing these gaps is critical for developing novel approaches to prevent infection and vaccine development.

Here we examine the initial phases of bradyzoite infection in mice and microfluidic models to simulate human intestinal interactions. By integrating microphysiological systems (MPS) with mouse infection studies, we aim to deepen our understanding of these processes in both mice and humans [34,35]. This study tracks early bradyzoite dynamics in the small intestine of orally infected mice and their replication in the jejunal stroma. We then use human MPS to explore the initial interactions and infections of bradyzoites in the human gut. Together, the use of both mouse and MPS allows us to identify key aspects of bradyzoite infection, including the site of replication in the villi, stage conversion kinetics, bradyzoite activation by pepsin, and parasite migration patterns within collagen. Our findings demonstrate that human MPS can effectively replicate bradyzoite conversion into tachyzoites and their infection patterns observed in mice. This work builds upon existing research and highlights the potential of MPS to study host-parasite interactions.

## Results

### *T. gondii* ingress to the intestinal stroma early in the infection

Studies have demonstrated that the jejunum is the primary intestinal niche of *T. gondii* growth in mice at day 6 post-oral infection [32]. To investigate the site of bradyzoite infection and eventual differentiation to tachyzoites, we analyzed the early time points of the infection in the jejunum of mice by following the natural route of infection through the consumption of brains containing cysts. To closely mimic natural conditions, we used the cystogenic M4 strain (type II strain), recently isolated from oocysts in our laboratory. This strain underwent minimal cell culture passaging, thereby more accurately representing a wild non-laboratory strain. The M4 strain exhibits a cystogenic phenotype in mice, providing a high yield of cysts per brain [36].

We infected a total of six mice across three independent experiments, each mouse fed two brains containing approximately $4 \times 10^2$ brain cysts of the M4 strain. To detect the parasites in the jejunal villi, we sacked the mice after 3 days post-feeding and scrapped the jejunal villi. We used polyclonal antibodies that recognize both stages of *T. gondii* due to the uncertain parasite stage at 3 days post-feeding, which precluded the use of stage-specific markers such as SAG1 or BAG1 antibodies. We employed actin staining to visualize the apical side of the enterocytes and villi structures (Figs 1A, 1B and S1). Although we collected numerous scraped villi from the jejunum of a total of six infected mice, detecting extracellular parasites or infected cells was challenging. We found single parasites in the jejunum, with diverse locations in the

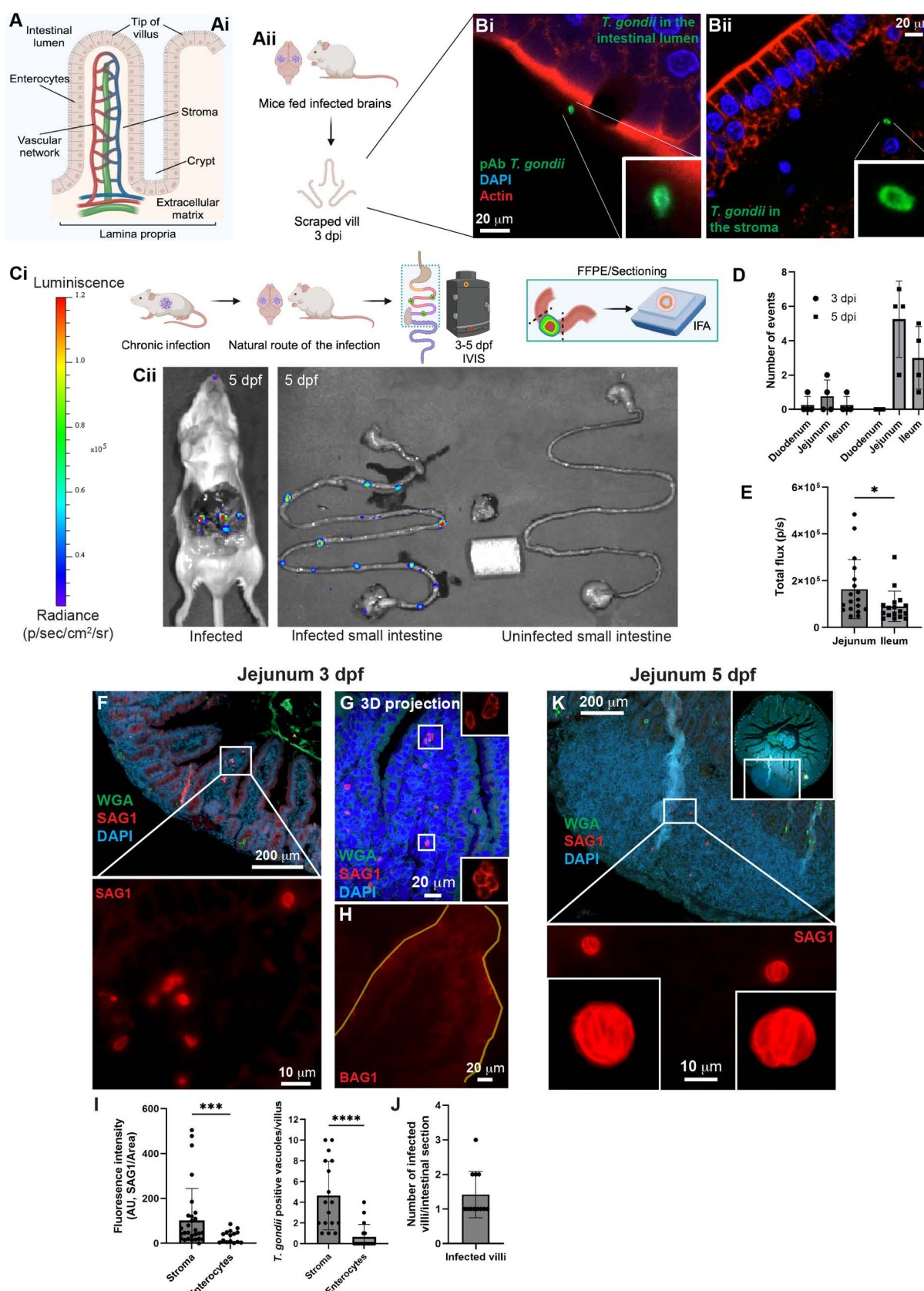

**Fig 1. *T. gondii* localization in mouse jejunal villi after ingesting cysts-containing brains. Jejunal localization of *T. gondii* after the natural route of infection.** (A) Schematic representation of (i) the villus and its structures and (ii) the experimental setup. Images represent single

events captured in independent infected mice after 3 days post-ingestion. Single mouse ingested brains containing ~$4 \times 10^2$ cysts. (B) Confocal images of infected villus with *T. gondii* after 3 days post-ingestion of cysts-containing brains. Each image represents single and isolated events captured in independent mice. *T. gondii* is localized in the intestinal lumen (i) or in the jejunal stroma (ii). In (B) the villi are stained for actin (red, rhodamine phalloidin), *T. gondii* (green, polyclonal antibodies), and nuclei (blue, DAPI). (Ci) Schematic representation of the experimental setup for detecting *T. gondii* infection by IVIS in the small intestine of mice fed cysts-containing brains. Single mouse ingested brains containing ~$4 \times 10^2$–~$5 \times 10^2$ cysts. (Cii) Representative image of the distribution of *T. gondii* infection by IVIS in the small intestine of mice fed cysts-containing brains after 5 days post-ingestion. (D) Grouped scatter plot showing the number of infection spots detected in the small intestine of seven mice in two independent experiments. (E) Grouped scatter plot of luminescence flux measured in the small intestine of infected mice. Data in (D) and (E) represent the observation of the events registered in seven infected mice in a combination of two independent experiments, with values presented as means ± SD (*$p \leq 0.05$). (F) Representative epifluorescent images of *T. gondii* infection in the jejunum of mice fed brain containing cysts at 3- and 5-days post-feeding. SAG1-positive parasites (tachyzoites) in the stroma. (G) Confocal 3D projection of tachyzoites replicating in the stroma. (H) Epifluorescent image showing the absence of BAG1 in a villus of the infected jejunum. Lines in yellow define the villus edges. (I) Grouped scatter plots showing the fluorescence intensity of SAG1 in infection spots (left plots) or the number of SAG1-positive PVs detected in either the stroma or enterocytes within the jejunum (right plots). (J) Quantification of the observed infected villi positive for SAG-1 parasites per section of intestine. (K) Representative of an epifluorescent image showing tachyzoites replicating within regions of excessive cell infiltration at 5 days post-feeding with brains. Intestines are stained for WGA (green), SAG1 (red), BAG1 (red), and nuclei (blue, DAPI). Data in (I) represents a combination of at least fifteen different intestinal structures, with values presented as means ± SD (***$p \leq 0.001$, ****$p \leq 0.0001$). Data in (J) represents a combination of at least 12 independent observations in different sections of the same or different infected mouse. Schematic representations in (A and C) were as created with BioRender.com.

villi. We observed parasites on the luminal side of the enterocytes (Figs 1Bi and S1B), inside an enterocyte as single parasites close to the nucleus (S1C Fig), near cell-to-cell junctions -suggesting transmigration between adjacent cells- (S1D Fig), in the stroma (Figs 1Bii, S1E and S1F); and, within a cell, most likely an immune cell which is characterized by its rounded shape and multi-lobed nuclei (S1G Fig). We did not see evidence of parasite replication in any of the 3-day samples. We observed infrequent foci of infection after examining hundreds of scraped villi from each infected mouse (S1 Fig). We could not quantify the findings, given that the number of observed events were limited.

## Bioluminescence imaging reveals dynamic distribution of *T. gondii* infection in mice intestines

Challenges in detecting parasites and infected cells in the scraped intestines included loss of villi directionality, villi/crypt integrity, and possible stroma damage due to the mechanical force when scraping. These issues necessitated the development of alternative methods to study early infection. We therefore infected mice with PruΔHPT:luciferase strain to obtain brain cysts. After day 28 post-infection, we isolated the brains and fed mice with brain containing cysts (Fig 1Ci). Each mouse ingested an approximate dose of $4 \times 10^2$–$5 \times 10^2$ brain cysts. Then, we used the *in vivo* imaging system (IVIS) to actively detect parasite bioluminescence following injection of luciferin [37,38], identifying infection sites in the peritoneal cavity of the mice and in their isolated small intestines at 3- and 5-days post-feeding (Figs 1Cii and S2).

To localize regions of infection, we euthanized the mice and exposed their peritoneal cavities to access the intestine (Figs 1Cii and S2). We detected *T. gondii* infection in the intestines of mice at 5 days post-feeding (Fig 1Cii). The distribution of the infection manifested as evenly distributed bioluminescent spots in mice along the small intestine (Figs 1Cii and S2). We detected poor to no bioluminescence signal in small intestines at 3 days post-feeding, suggesting that the infection was still in the early stages with little to no replication (S2B and S2D Fig). At 5 days post-feeding, intestines showed infected areas predominantly in the jejunum (Fig 1D and 1E). After 5 days post-feeding, the infection levels in jejuna were higher than those observed in ilea (Fig 1D and 1E). No signal was detected in food and a piece of muscle isolated from the leg of infected mice, both used as a control (Figs 1Cii and S2). Upon isolating the mouse intestines, we observed enlarged lesions that were visible to the naked eye and

colocalized with the bioluminescent infection foci detected by IVIS (S2B Fig). These lesions were consistently identifiable in all infected mice at 5 days post-infection and were absent in the intestines of uninfected controls.

## Parasite distribution and replication dynamics in mouse intestinal tissues

To investigate parasite replication and stage conversion, we collected the bioluminescent areas in the jejunum and ileum at 3 and 5 days-post feeding for paraffin embedding (represented in Fig 1Ci). We stained the sections with antibodies against SAG1 and BAG1 to determine the parasite stages (Figs 1F, 1G, 1H, 1K, S3 and S4), or with polyclonal antibodies against *T. gondii* for a general overview of the infection (S5 and S4 Figs). Of note, we used wheat germ agglutinin to stain the intestinal membranes, as actin staining, which worked well for fresh or fixed villi, was ineffective for paraffin-embedded intestinal sections, possibly due to the long dehydration time during sample processing.

At 3 days post-feeding, we only observed SAG1-positive parasites in the jejunal stroma with the absence of BAG1-positive parasites (Figs 1F, 1G, 1H, S3A and S4A) and within the enterocytes (Fig 1F, 1G and 1H). Additionally, we detected single parasites or small parasitophorous vacuoles (PVs) near the muscular layer beneath the crypts (S4A Fig). Parasites located in the stroma were replicating, as evidenced by vacuoles containing more than one parasite (Figs 1G, S4B, S4C and S1 Movie). Scarce parasites were found replicating in the enterocytes (Fig 1I). Despite the ingestion of hundreds of cysts by the mice, infection foci were confined only to one or two villi per intestinal section in more than 95% of the sections analyzed (Fig 1J). We found evidence of parasites in the lamina propria and inside a rounded cell, which most likely are immune cells (S4A Fig). For the rest of the parasites, we cannot state whether they replicated in immune cells or in intestinal cells (S4A Fig).

At 5 days post-feeding, we noted a prominent area full of cells between the crypts and the muscular layer (Figs 1K and S5A) and detected SAG1-positive/BAG1-negative parasites replicating in the area (Figs 1K, S3B and S2 Movie). We confirmed the presence of immune cells in the bumped areas by staining against CD45 (S5B Fig). The enlarged areas were detected by the naked eye, colocalized with bioluminescent areas seen by IVIS and present in all the intestinal sections analyzed at 5 days post-ingestion. Hundred percent of the bumps were infected or surrounded by parasites. Our samples showed CD45-positive cells adjacent to infected host cells (S5B Fig); however, infected cells were CD45-negative which suggests that tachyzoite's replication continues in intestinal cells even after 5 days-feeding (S5C Fig). CD45-positive cells were also found in uninfected intestines close to the crypts and in the villus (S5E Fig). We did not find any cell aggregate structures in uninfected intestines (S5A Fig), and we detected parasites close to endothelial tissue containing red blood cells (S5D Fig). An anti-CD31 antibody-stained erythrocytes (S5F Fig), but not the capillary network in the intestines (S2G and S5F Figs).

## Optimization of culture conditions for bradyzoite differentiation

*In vitro* cultured bradyzoites have proven highly beneficial for studying bradyzoite biology, particularly due to the reduction in time and animals [39]. Our laboratory commonly grows tachyzoites at 33°C to slow parasite growth and reduce the number of parasite passages required for maintenance. We took advantage of slow replicating tachyzoites to induce bradyzoite formation using the widely used alkaline pH differentiation media [39]. The new differentiation condition includes culturing tachyzoites in media with a pH 8.1 at 33°C [39], in a thermostatic incubator under normoxic conditions with ambient $CO_2$ concentrations. We followed parasite replication in standard media (HFF media) and differentiation conditions

using ME49 mCherry parasites at 33°C. ME49 mCherry parasites grown in HFF media lysed the host cells by day 8 (Figs 2A and S6A). In contrast, in the presence of differentiation media, parasites remained still intracellular after 8 days (Figs 2A and S6A).

Temperature played a large role in bradyzoite *in vitro* differentiation. ME49 parasites differentiated at 37°C only yielded ~30% of BAG1 positive parasites after 6 days in differentiation media; in contrast, almost a complete differentiation to bradyzoites (>95% BAG1 positive) were obtained by 33°C incubation in differentiation media (Figs 2B and S6, BAG1/SAG1). Our new switching conditions yielded a stable high efficiency of bradyzoites differentiation (>97%) up to 12 days post-switching (Fig 2B, right graph). *Dolichos biflorus* agglutinin (DBA) decorated the cyst wall in 98% of ME49 parasites as soon as 7 days in differentiation media (Fig 2C) and remained still DBA-positive at 12 days in differentiation media (Fig 2D). Our conditions achieved 92–98% cysts formation in type II strains Pru and ME49 (Fig 2D). RH parasites rarely expressed BAG1 in pH 8.1 media at 37°C followed by rapid cell lysis (Fig 2B, left graph), but reduction of the temperature to 33°C generated ~90% of DBA positive vacuoles in RH parasites (Figs 2C and S6D).

To investigate real-time switching dynamics, we used the type I/III hybrid strain EGS LDH2p-GFP/SAG1p-mCherry parasites [40] and measured the expression of stage-specific fluorescent reporters to track the point of stage conversion (Fig 2E). While incubation of EGS at 37°C in differentiation media led to cell lysis after 3–4 days post-infection (Figs 2E and S7A), at 33°C complete bradyzoite differentiation occurred within 6 days in differentiation media, as evidenced by the presence of GFP/LDH2-positive cysts structures and the absence of SAG1/mCherry-positive parasites (Figs 2E and S7B). After 9 days in differentiation media, we detected free floating GFP-positive cyst structures, possibly due to host cells detaching from the substrate or cyst structures being expelled from the host cells (S7 Fig, 9 dpi). Given the consistent bradyzoite development across strains, we decided to harvest *in vitro* cysts between 7–9 days in differentiation media for all experimental procedures.

## Evaluation of *in vitro* bradyzoites in host systems

*In vitro* bradyzoites differentiated at 33°C in switching media exhibited molecular and physiological features similar to brain cysts, such as: positive for DBA and BAG1, negative for SAG1, rounded shape structures, slow replication parasites, and resistance to pepsin digestion (addressed below). ME49 mCherry *in vitro* bradyzoites infected and replicated in both HFF and Caco-2 cells, indicating their viability after incubation at 33°C (S6F Fig). To assess the virulence of the *in vitro* cysts, we fed Swiss Webster mice with *in vitro* cysts on bread, mimicking the natural route of infection. One mouse fed ~$1 \times 10^4$ and three mice fed ~$7 \times 10^4$ ME49 mCherry *in vitro*-differentiated cysts developed chronic infection after 28 days, as evidenced by cyst isolated from the brain (S6G Fig). In contrast, mice fed ~$1 \times 10^4$ EGS *in vitro* cysts did not exhibit brain cysts formation or succumbed to the infection after 28 days post-feeding (Fig 2G). To evaluate the virulence of EGS *in vitro* cysts further, three mice per group were fed with EGS *in vitro* cysts ranging from ~$1 \times 10^2$ to ~$1 \times 10^5$. Only mice fed with ~$1 \times 10^5$ *in vitro* cysts succumbed to the infection after 11–13 days (Fig 2G). Brain cysts were not detected in the remaining EGS-infected mice, likely due to the nature of this type I/III hybrid strain.

## Bradyzoite activation depends on pepsin digestion

Our group has successfully used the microphysiological system (MPS) and Caco-2 cells to model *T. gondii* tachyzoite-specific interactions during the infection and study the innate immune response [35]. We turned to these devices to model early intestinal bradyzoite infection using Caco-2. MPS previously demonstrated confluent distribution throughout the

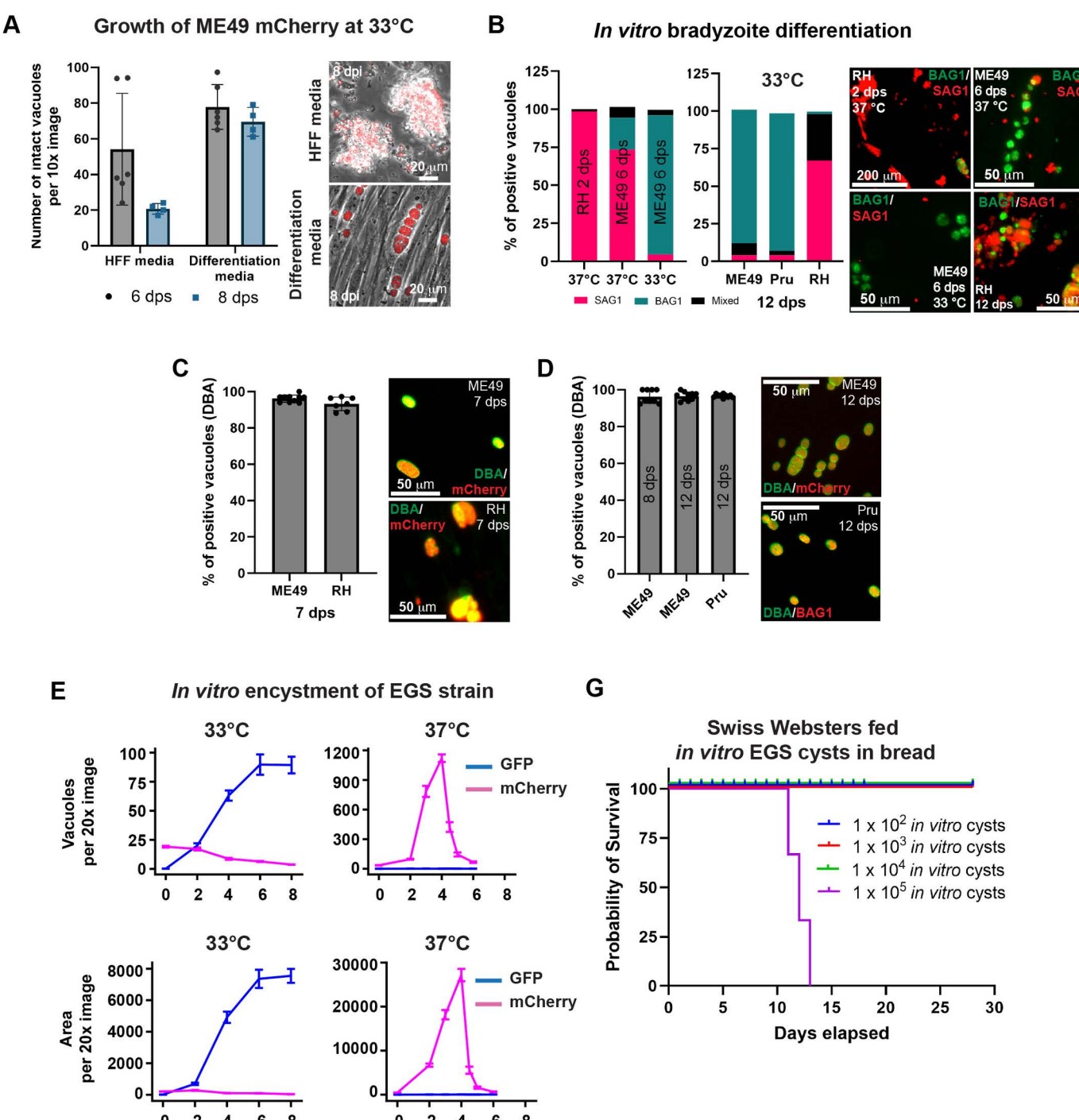

**Fig 2. *In vitro* bradyzoite differentiation of type I and II strains at 33°C.** (A) Plot represents the number of intact vacuoles of ME49 parasites in the presence of HFF media or differentiation media. Quantification was done in at least ten different fields after 6- and 8-days post infection. Images to the right show representative fluorescent imaging of infected HFF cells. (B) Plots to the left shows the quantification of positive vacuoles to SAG1, BAG1 or both in ME49 parasites after incubation for 6 days in differentiation media at 37°C or 33°C. Plots to the right show positive vacuoles to SAG1, BAG1 or both in type II and type I parasites after incubation for 12 days in differentiation media at 33°C. Images to the right show representative fluorescent imaging of infected HFF cells. (C) Plots to the left show the quantification of PVs positive to DBA after 7 days in differentiation media in ME49 and RH parasites. Images to the right show representative fluorescent images of DBA parasites (D) Plots to the left show the quantification of PVs positive to DBA after 8- and 12-days in differentiation media in ME49 and RH parasites. Images to the right show representative fluorescent images of DBA- or BAG1-positive type I parasites after 8- or 12-days in differentiation media. (E) Plots show the i*n vitro* differentiation of EGS strain at 33°C and at 37°C. Plots show live fluorescent imaging of EGS parasites detecting mCherry-expressing tachyzoites or GFP-expressing bradyzoites. Quantifications were

performed by quantifying the number of vacuoles (top plots) or by measuring the fluorescent area (bottom plots). (G) Survival curve of mice fed bread with a range of $1 \times 10^2 - 1 \times 10^5$ in vitro-generated bradyzoites. Three Swiss Webster mice were tested for each experimental condition.

lumen and along with the presence of mucus-producing goblet cells marker MUC2A, the tight junction protein ZO-1 (confirming the stability of the cells forming the lumen), and the microvilli marker villin [35]. We confirmed the homogeneous and close distribution of Caco-2 cells in the current study by actin staining (Fig 3Ai). Barrier functionality and luminal coverage were validated by the maintenance of FITC-dextran within the lumen (Fig 3Aii). We tested the viability of Caco-2 cells in the MPS by infecting them with RH mCherry tachyzoites and detecting active replication after 48 hours post-infection (Fig 3Aiii).

We infected Caco-2 lumens with *in vitro*-differentiated ME49 mCherry or EGS bradyzoites to study the early points of the infection. We simulated the oral infection by degrading the cyst structures with pepsin from porcine stomach for 2 min to exteriorize bradyzoites; however, we did not detect replicating parasites in any of our lumens after 5 or more days of infection (S8 and S9 Figs). This result was unexpected because bradyzoites from the same sample successfully infected and replicated in Caco-2 monolayers (S6B Fig). We confirmed that the *in vitro* bradyzoites were viable after 2 minutes of pepsin treatment through SYTOX green with 97% being viable (S8C Fig). We assessed Caco-2 cells viability and accessibility to *T. gondii* in lumens by infecting them with ME49 mCherry tachyzoites and observed replication after 3 days post-infection (S8D Fig), suggesting the defect in replication was specific to bradyzoites. We hypothesized that bradyzoites need to be activated by extending the treatment time with pepsin, as recently suggested [41].

We tested the role of pepsin in bradyzoite activation in 2D monolayers. We incubated ME49 mCherry brain cysts with pepsin from 2 min up to 2 h, and then monitored parasite replication over 9 days in HFF monolayers (Fig 3B). As a negative control of pepsin activation, we mechanically disrupted cysts by passing them through a 30-gauge needle. Bradyzoites stimulated with pepsin for 30 min displayed the highest replication, followed by those stimulated for 15 min and 1 h, which exhibited similar replication patterns (Fig 3B). Bradyzoites exposed to pepsin for 2 minutes and 2 hours showed lower levels of parasite replication (Fig 3B) as well as syringe-lysed cysts (Figs 3B and S10A). *In vitro* differentiated EGS bradyzoites were also effectively activated by incubating with pepsin for 30 minutes, compared to 2 minutes or syringe lysis (S10B Fig, top graph).

## Optimization of human cell lines for use in microphysiological systems

Given that our MPS are seeded with Caco-2 cells, we ensured that bradyzoites infect Caco-2 cells under our new activation conditions. We compared 30 min pepsin-activated brain bradyzoite infection in Caco-2 vs HFF monolayers. HFF cells exhibited higher susceptibility to bradyzoite infection compared to a limited infection observed in Caco-2 cells (Figs 3C, S9A, S3 and S4 Movies). We then tested the human colon epithelial cell line (HIEC-6), which displayed high susceptibility to bradyzoite infection similar to HFF cells (S10A Fig and S5 Movie). However, the HIEC-6 cells both did not form uniform lumens in the MPS and migrated outside the lumen into the collagen I/fibronectin matrix (S6 Movie). As such, we used Caco-2 cells for generating human-intestine lumens, as they are more effective than HIEC-6 at accurately modeling luminal structures and maintaining epithelial integrity.

## Modeling intestinal bradyzoite infection using human intestinal microphysiological systems

We examined bradyzoite-to-tachyzoite conversion in human intestinal MPS, by infecting Caco-2 lumens with brain-isolated ME49 mCherry bradyzoites (Fig 4A). At 3 days

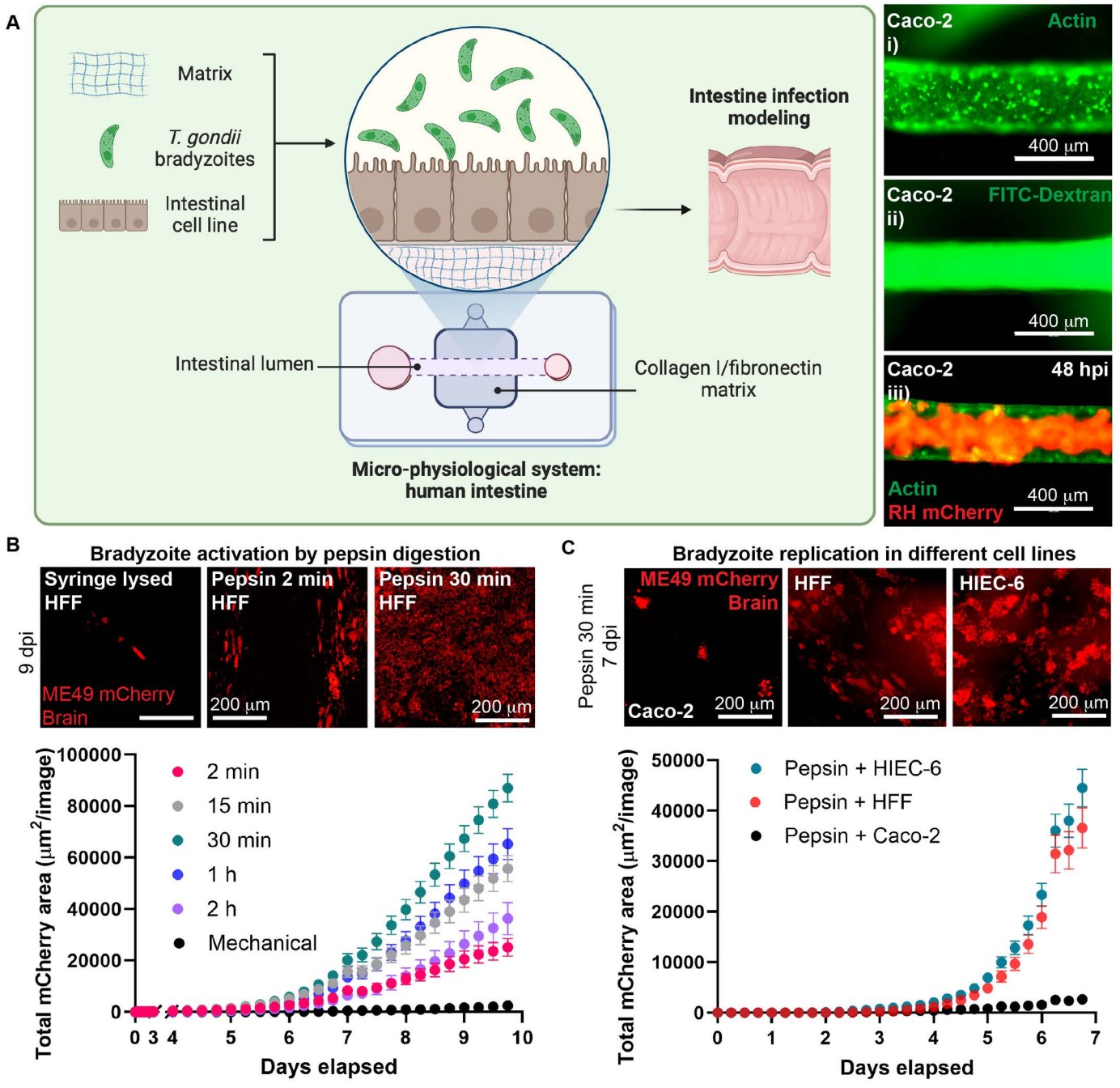

**Fig 3. Bradyzoite activation for infection of human intestinal microphysiological systems to study bradyzoite interactions.** (A) Schematic representation of bradyzoite infection niche in human intestinal MPS. (i) Live staining of actin showing the coverage of the lumen by Caco-2 cells. (ii) FITC-dextran live imaging assessing the polarity of the Caco-2 cell line in the MPS. (iii) Parasite replication at 48 hours post-infection with $1 \times 10^3$ RH tachyzoites as a control for host cell viability. (B) Representative fluorescent images of parasite replication in HFF cells infected with brain bradyzoites, either activated by pepsin or subjected to mechanical lysis, at 9 days post-infection. Graph shows parasite kinetics of growth in HFF for pepsin-activated versus inactivated bradyzoites. (C) Representative fluorescent images of replication in HFF, Caco-2, and HIEC-6 infected with activated bradyzoites at 7 days post-infection. Graph shows parasite kinetics of growth in the three different cell lines. Schematic representation in (A) was created with BioRender.com.

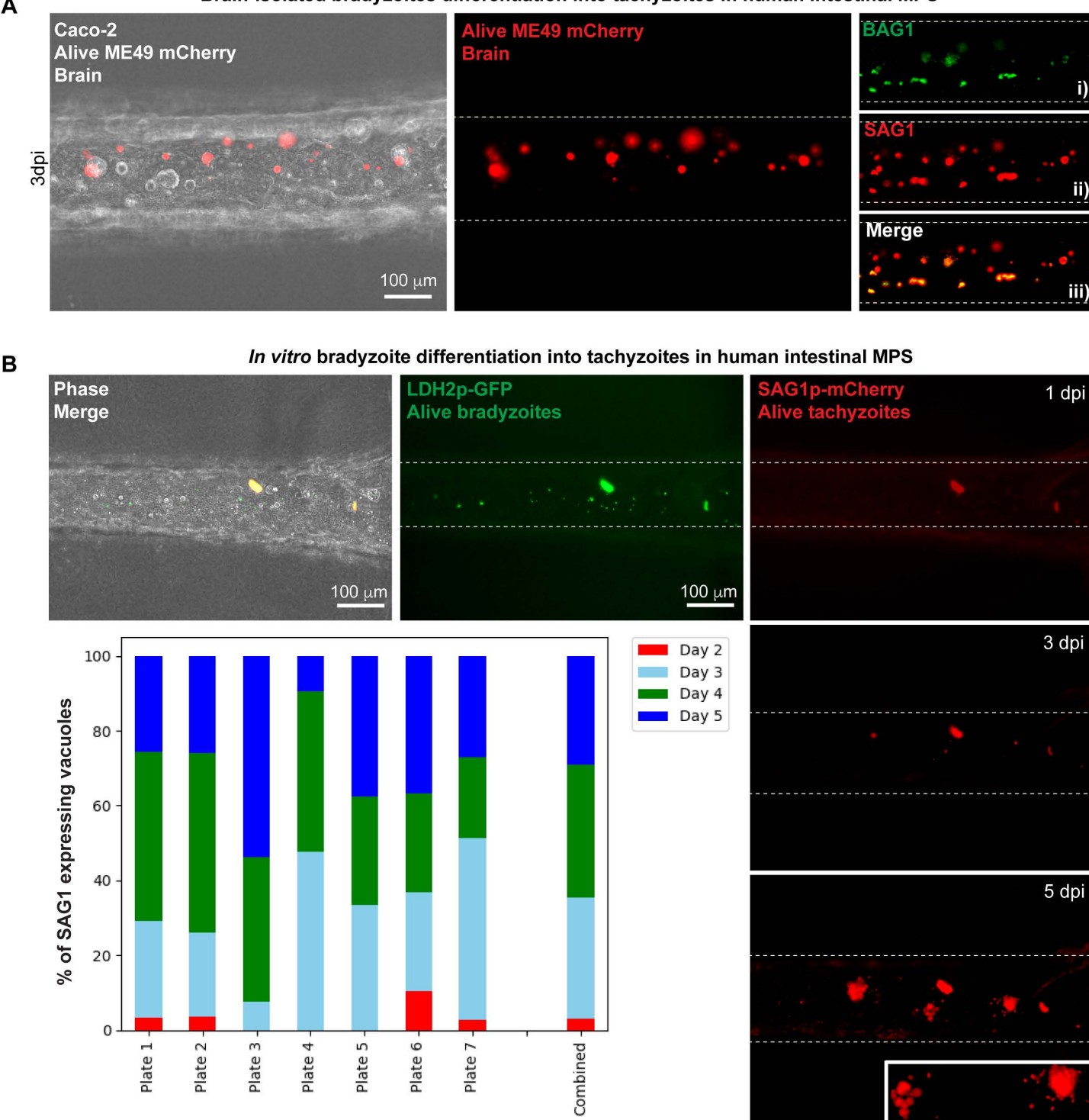

**Fig 4. Bradyzoite differentiation to tachyzoite in human intestinal microphysiological systems.** (A) Fluorescent images showing replication of alive ME49 mCherry bradyzoites in the Caco-2 lumen. (i)(ii), and (iii) show immunofluorescent images of the parasite conversion of ME49 mCherry bradyzoites to tachyzoites, with markers for tachyzoites (SAG1, red) and bradyzoites (BAG1, green) at 3 days post-infection. (B) Fluorescent images of alive EGS LDH2p-GFP/SAG1p-mCherry parasites undergoing differentiation in MPS, with bradyzoites shown in green and tachyzoites in red. Punctuated lines represent the edges of the lumen and its boundary with the matrix. Lumens in (A) and (B) were infected with a total of $3 \times 10^4$ bradyzoites ME49 mCherry or EGS LDH2p-GFP/SAG1p-mCherry parasites. Plots represent the quantification of mCherry positive PVs (expressing SAG1) from day 2- to 5-post infection as bradyzoites. Plots are graphed as the average of positive PVs per lumen. A total of seven plates were analyzed, each one with at least three infected lumens.

post-infection, parasites were fixed, mCherry fluorescence was quenched using methanol (S11 Fig), and then parasites were stained with SAG1 and BAG1 antibodies (Figs 4Ai-iii). Although most PVs were SAG1-positive, we also detected BAG1-positive PVs (Fig 4Ai). Notably, BAG1-positive PVs were often co-labeled with SAG1, indicating that they were converting to tachyzoites (Fig 4Ai-iii).

To monitor stage conversion in real-time, we infected Caco-2 lumens with *in vitro* bradyzoites of the EGS strain and quantified tachyzoite conversion by counting red PVs (mCherry/SAG1 expressing parasites). At 2 days post-infection, > 90% of PVs contained GFP-expressing bradyzoites (Fig 4B). Consistent with the ME49 strain, tachyzoite conversion was clearly detected by day 3 post-infection in 30% of vacuoles (Fig 4B). By day 4 post-infection, tachyzoites were present in 70% of PVs, and near complete tachyzoite conversion was evident by 5 days post-infection with only mCherry-expressing EGS parasites detected (Fig 4B). These findings mirror *in vivo* results where tachyzoites were detected in mouse intestinal sections at 3 days post-feeding (Fig 1). Our 2D models using HFF cells suggest that tachyzoite conversion from EGS bradyzoites occurs after approximately four replication cycles, as evidenced by the following the same PV for up to 4 days. Rapid tachyzoite replication was observed after 3 days post-infection, as shown in our Incucyte data obtained (S10C Fig).

## Bradyzoites rapidly transmigrate the intestinal barrier without replicating in the enterocytes

During the early stages of bradyzoite infection in the MPS, we observed parasites localizing to the collagen I/fibronectin matrix within 20 hours post-infection. This finding suggests that bradyzoites may rapidly traverse the Caco-2 cell layer within the lumens (Fig 5A). To ensure the observed transmigration behavior reflects realistic biological processes *in vivo* and to confirm that these observations are not artifacts of the MPS system but reflect true bradyzoite behavior, we isolated mouse jejunal villi and infected them with *in vitro*-differentiated EGS bradyzoites. Our observational results showed that bradyzoites localize within the stroma in villi after 6 or 24 h post-infection (Fig 5Bi and 5Bii). We also observed bradyzoites migrating between enterocytes, indicating that the parasites may reach the stroma without necessarily infecting the enterocytes themselves (Fig 5Biii and 5Biv).

To avoid the structural villi damage when scraping previously discussed (Figs 1 and S12), we opted to use fixed villi prior to bradyzoite infection. In fixed villi, we detected bradyzoites beneath the enterocytes after 6 h post-infection (Fig 5Ci) and an active bradyzoite internalization into the stroma (Fig 5Cii). As expected, we did not see bradyzoites within the cells, indicating that they cannot invade the fixed tissue. Consistent with observations in unfixed villi (Fig 5Biii), we found bradyzoites adjacent to areas of open enterocytes (Fig 5Ciii), suggesting a process of transient aperture of the enterocytes to reach the stroma (Fig 5iv).

## Tachyzoites can migrate collagen and infect surrounding cells

In our intestinal sections, we observed both replicating and individual parasites within the stroma (Figs 1 and S1), but it was still unclear whether parasites migrated by entering immune cells or as extracellular parasites crossing biological barriers. The potential for migration in immune cells to reach the capillaries and organs appears highly feasible [42–45], but there is also evidence suggesting active parasite migration through the tissues [32]. To investigate whether parasites can disseminate without the aid of a host immune cell, we used an MPS with 2 lumens, each of which was seeded with Caco-2 cells with one lumen infected with *in vitro* EGS bradyzoites (Fig 6A) and the other left uninfected. We monitored the behavior of parasites exteriorized from Caco-2 cells throughout the experiment every 24 h. By 3

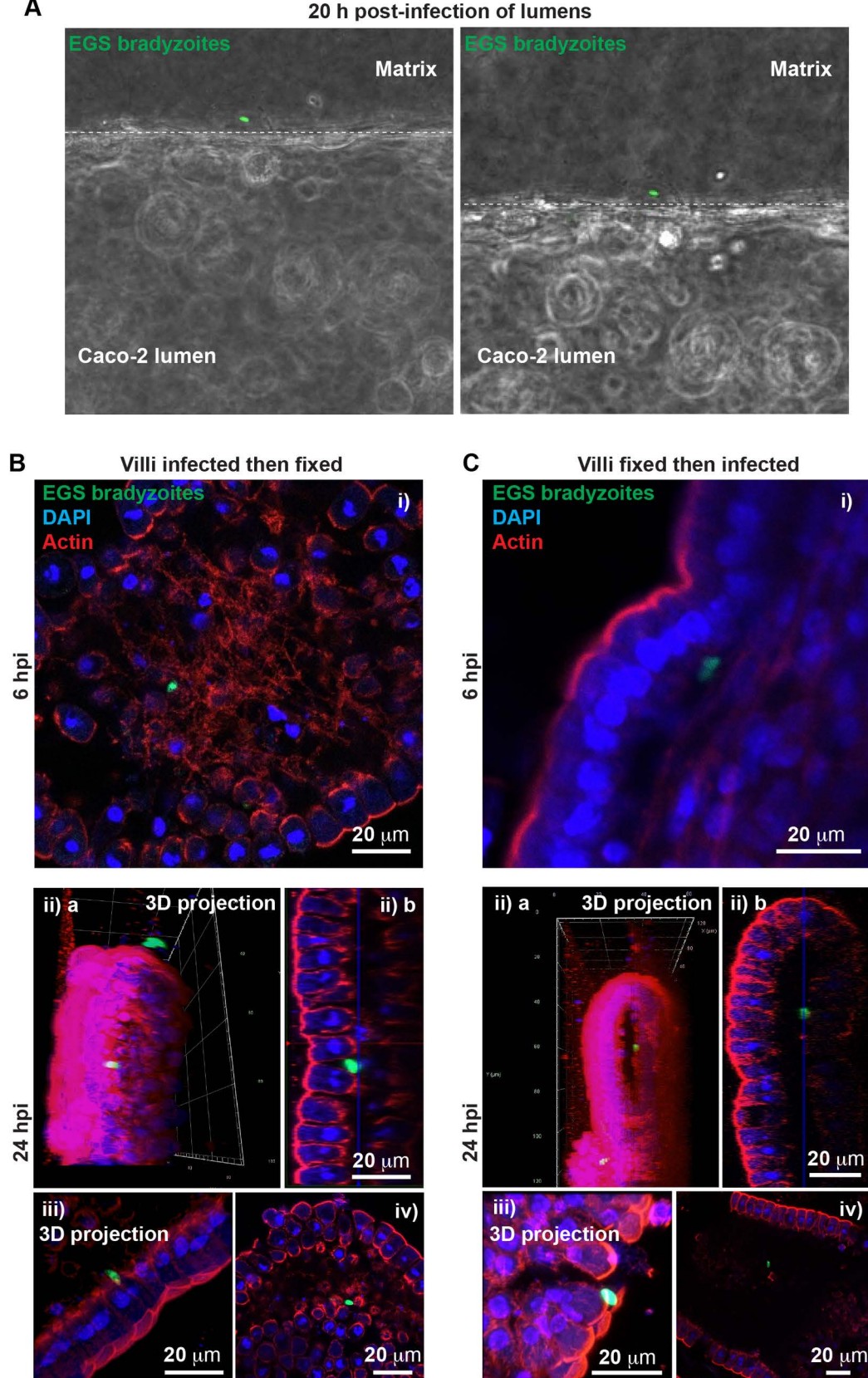

**Fig 5. Bradyzoite transmigration of Caco-2 cells in human intestinal microphysiological systems and intestinal jejunal villi explants.** (A) Representative fluorescent images showing bradyzoite transmigration through the Caco-2 cells after 20 h

post-infection of lumens. Punctuated lines in white represent the edges of the lumen and its boundary with the matrix. (B) Confocal images of explants of villi infected for 6 h (i) or 24 h (ii-iv) and then fixed. (i) Shows EGS LDH2p-GFP/SAG1p-mCherry bradyzoites localizing in the jejunal stroma. (ii a) shows a 3D projection of a bradyzoite localizing in the stroma of villus. (ii b) represents a Z-section of (ii a). (iii) Shows a 3D projection of bradyzoites transmigrating through the enterocytes. (iv) Confocal image of a bradyzoite localizing in the stroma of villus. (C) Confocal images of explants of villi fixed and then infected for 6 h (i) or 24 h (ii-iv). (i) Shows EGS LDH2p-GFP/SAG1p-mCherry bradyzoites localizing underneath the enterocytes. (ii a) shows a 3D projection of a bradyzoite localizing in the stroma of the infected fixed villus. (ii b) represents a Z-section of (ii a). (iii) 3D projection of bradyzoites interacting with the enterocytes of the fixed villus. (iv) Confocal image of a bradyzoite localizing in the stroma of fixed villus.

days post-infection, a mixture of tachyzoites and bradyzoites were detected in the lumen, with additional free bradyzoites observed in the collagen I/fibronectin matrix (Fig 6B). By 5 days post-infection, tachyzoites had migrated through the collagen I/fibronectin matrix and reached the adjacent uninfected Caco-2 lumen (Fig 6C). By 6 days post-infection, we observed replicating parasites in the uninfected lumen, indicating the ongoing expansion of the infection *in vitro* (Fig 6D).

## Discussion

*T. gondii* enters the host through ingestion, with the intestine as the primary site of infection initiation. Studying the bradyzoite's first interaction within the intestine has been limited given the complexity of working with mouse intestine and the inadequacies of live imaging sensitivity [38]. While previous works have explored the interaction of tachyzoites with human intestinal cell lines in 2D models [46], studies on bradyzoite infections in these cell lines are notably scarce. To fill these gaps, we used MPS with human intestinal cells to complement our examination of bradyzoite infection in a mouse model. We examined the dynamics of host-parasite interactions such as ingress, site preferences for invasion and replication, and the kinetics of stage conversion from bradyzoites to tachyzoites within the gut.

Although the initial site of the infection has been controversial between the jejunum [32,33] and the ileum [29,30,33], our findings support *T. gondii* localization within both sections during early infection with a preference for parasite replication in the jejunum. For that reason, we focused on analyzing the jejunum for our *in vivo* infections. Parasites were predominantly observed in the intestinal stroma, with fewer parasites in enterocytes (Fig 1). Particularly noteworthy was the early presence of parasites near cell-to-cell junctions, suggesting active transmigration from the lumen to facilitate dissemination [30,32]. Previous research has suggested that extracellular tachyzoites degrade tight junction proteins and affect the epithelial polarity in 2D and 3D-transwell models as a mechanism for parasite active transmigration of the biological barriers [47–49].

In contrast to previous reports where intestinal infection often concentrated at the villi tips [33], our study found limited replication within enterocytes (Figs 1, S1, S3 and S4). This result may be attributed to the rapid turnover of enterocytes as in humans and mice, they are typically replaced every 2 to 3 days [50]. For sustained replication and dissemination, *T. gondii* may target more stable environments such as stromal tissue. Previous work has shown sporozoites in the lamina propria after just 6 hours post-oral ingestion of oocysts [30], suggesting a rapid and transient migration through the enterocytes. Parasite avoidance of enterocytes may also be driven by unfavorable metabolic activity in enterocytes or exposure to digestive enzymes in enterocytes [51].

In previous reports, sporozoite or bradyzoite conversion to tachyzoites relies on qualitative assessments using transmission electron microscopy or embedded tissue sections, lacking stage-specific antibody staining [29–31]. In our study, we used SAG1 for tachyzoites and

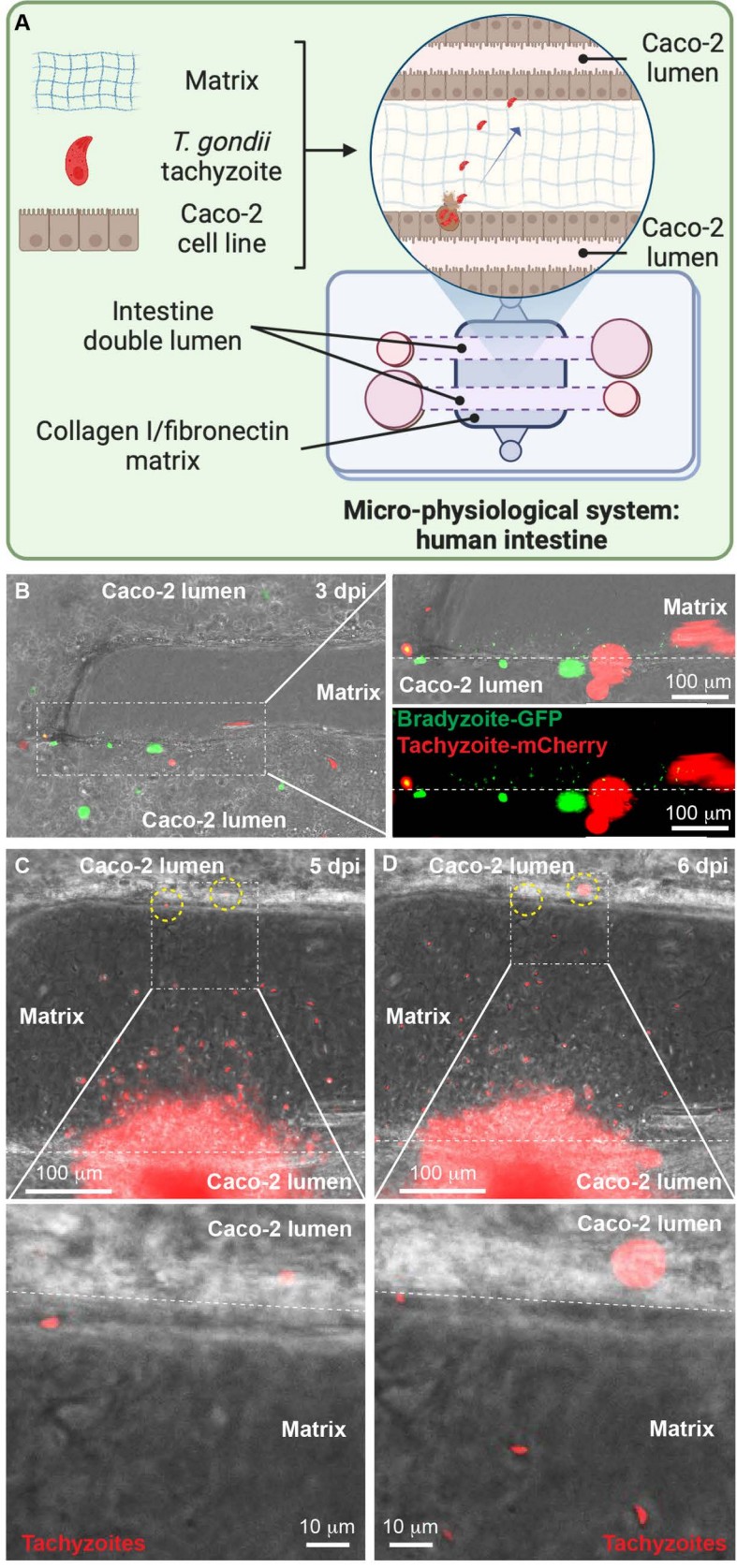

**Fig 6. Active tachyzoite migration through 3D collagen I/fibronectin matrix and expansion of its infection in human intestinal MPS generated with Caco-2 double-lumens.** (A) Schematic representation of bradyzoite/

tachyzoite infection niche in human intestinal MPS. (B) Fluorescent images showing the coexistence of EGS LDH2p-GFP/SAG1p-mCherry bradyzoites and tachyzoites in the infected intestinal lumen at 3 days post-infection, with insets highlighting the localization of bradyzoites in the 3D collagen I/fibronectin matrix. Fluorescent images tracking the migration in collagen and Caco-2 infection by EGS LDH2p-GFP/SAG1p-mCherry parasites after exteriorization from Caco-2 cells in the bottom Caco-2 lumen shown in (C) and (D). (C) Micrographs show tachyzoites localizing in the 3D collagen I/fibronectin matrix and actively migrating towards the top Caco-2 lumen by day 5 post-infection. (D) Right micrographs show tachyzoites expanding the infection to the adjacent top Caco-2 lumen. Punctuated circles show parasitic infection to the top Caco-2 lumen. Punctuated lines in white represent the edges of the lumen and its boundary with the matrix. Lumens were infected with $5 \times 10^4$ bradyzoites of the EGS LDH2p-GFP/SAG1p-mCherry strain. Magnified photograph in (B) shows different parasites captured in different planes than those in unmagnified photograph. Prominent areas in (C) and (D) are the result from foci of infection. Schematic representation in (A) was created with BioRender.com.

BAG1 for bradyzoites to follow their conversion process, as these stage specific markers are commonly used in the field [52]. Tachyzoites exhibited active replication within the stroma starting at day 3. Detection of parasites replicating in the jejunum before day 3 post-ingestion was not feasible in our samples using embedded tissue or *in vivo* imaging, likely due to the initially low number of parasites on 1- and 2-days post-feeding. This limitation prompted us to use intestine scrapes and subsequently transition to MPS for our investigations (Figs 4–6). The formation of slightly swollen intestinal areas by day 5 post-ingestion, possibly due to infiltration of immune cells, further underscores the complex interplay between *T. gondii* and the host immune response [32,33]. The observation that enlarged and infected intestinal areas were visible to the naked eye suggests the potential to identify foci of infection without the need for a fluorescent strain. However, this phenomenon requires further investigation and quantification, which will be addressed in a subsequent report. Parasites were found near endothelial tissues (S5D Fig), suggesting potential routes for hematogenous spread, aligning with observations of Trojan horse-like migration and their infiltration to the central nervous system [45].

*In vitro* differentiated bradyzoites have been used to progress the understanding of HFF cell infection dynamics, particularly the role of the moving junction [53]. *In vitro* bradyzoites were recently evaluated for their structural, metabolic, and functional similarities to *in vivo* cysts, resembling them in ultrastructure, temperature stress, and tolerance to antiparasitic agents. They also presented functional hallmarks to *in vivo* cysts infectivity in mice and their resistance to pepsin digestion for up to 60 minutes [54]. These findings provided a foundation for using *in vitro* bradyzoites as a model to investigate bradyzoite infections. Our revised protocol for cultivating bradyzoites *in vitro* yielded high quantities for type II strains and effectively induced bradyzoite markers in the poorly cystogenic RH strain (Fig 2). The resistance to pepsin digestion was crucial for activation, confirming their functionality as bradyzoites. Pepsin digestion has been extensively used for exteriorizing *T. gondii* cysts in a period of time between 2 min and 2 h [53–57]. Currently, there is a published thesis showing pepsin treatment activates bradyzoite invasion [41]. In our study, we found that pepsin digestion plays a crucial role in activating *T. gondii in vitro* cultured- and brain isolated-bradyzoites, leading to enhanced parasite replication (Fig 3). Pepsin digestion was also vital to remove any remaining tachyzoite in our *in vitro* bradyzoite cultures, which agrees with previous findings [58,59].

Using human fibroblasts infected with brain cyst bradyzoites, researchers have previously observed the coexistence of bradyzoite and tachyzoite markers initially, with exclusive SAG1 expression emerging after 48 hours post-infection [59]. However, understanding this process in biologically relevant tissues such as the intestine remains incomplete. To address the gaps in understanding bradyzoite-to-tachyzoite conversion, we used Caco-2 cells in MPS mimicking human intestines (Fig 4). Bradyzoite infection and replication were significantly lower in

Caco-2 cells compared to HIEC-6 (Fig 3C). Both lines are derived from cancer patients, but Caco-2 cells are derived from the colon, and HIEC-6 cells are derived from the small intestine [60–62]. In mice, the small intestine has been recognized as the predominant site of *T. gondii* infection [32]. In the MPS, HIEC-6 cells displayed unusual behavior by both failing to adhere to the collagen to form a complete lumen and extravagating into the collagen I/fibronectin matrix (S6 Movie). These phenotypes may be due to HIEC-6 inability to form tight junctions [63]. Caco-2 cells excelled in forming complete circular lumens in our MPS and consistently showed tachyzoite conversion within 3 days post-infection aligning with our *in vivo* results (Fig 4).

Future studies incorporating co-cultures with gut microorganisms or immune cells in the human intestinal MPS could enhance our understanding of bradyzoite switching, potentially accelerating the process. Recently, researchers have published the use of endothelial cell lines in MPS for *T. gondii* tachyzoites studies opening the avenue for studying a complete bradyzoite and tachyzoite dissemination in the presence of immune cells [35,64]. These models offer valuable insights into stage conversion and infection progression, complementing traditional in vivo approaches. Additionally, co-infections with parasites like *Schistosoma, Giardia*, or *Entamoeba histolytica*—common in high-parasitic regions—may influence *T. gondii* virulence. The complex interactions between these pathogens and the immune response could impact *T. gondii* motility, switching behavior, and disease outcomes, making co-infection studies in MPS models crucial for understanding these dynamics.

Our results demonstrate that bradyzoites actively migrate through the enterocyte layer and localize within the stroma, reminiscent of earlier studies where tachyzoites were observed in the lamina propria following oral ingestion of cysts or oocysts [28–30]. In the absence of systemic immune cells in our villi explants, this migration indicates that bradyzoites can actively penetrate the lamina propria. The parasites' movement through the epithelial layer underscores their capability to invade deeper in the intestine early in the infection process. These findings validate the hypothesis that parasites actively traverse epithelial layers for dissemination purposes, potentially facilitated by effectors secreted by the parasites as previously suggested [47,65].

Tachyzoites also exhibited active migration through the collagen I/fibronectin matrix, exceeding 200 μm in width, to reach and infect adjacent Caco-2 lumens (Fig 6). This phenomenon mimics potential host cell-independent routes of *T. gondii* dissemination within the host as a possible route for reaching distant intestinal areas. Using collagen I for the lumens was particularly beneficial for tracking parasite migration, aligning with earlier studies using 3D collagen matrix assays to study the motility of tachyzoite-infected dendritic cells [66]. Other studies using MPS have employed collagen I as a matrix to study endothelial parasite migration [35,64]. Exploring alternative substrates, such as other types of collagens, laminin, or Matrigel, all of which have been used as 3D models for studying tachyzoite motility [67–69], could provide deeper insights into *T. gondii* migration, motility, host cell invasion, and infection dynamics in MPS.

MPS offer a promising platform to investigate interactions between bradyzoites and immune cells such as neutrophils and macrophages, which have traditionally been studied in 2D models. Our platform also provides the foundation for modeling human organs, such as the placenta for studying congenital toxoplasmosis, the eye for studying chorioretinitis, and the brain to study infection of the CNS and encephalitis. This study elucidates critical aspects of *T. gondii* early infection dynamics, including localization, differentiation, and migration within intestinal tissues. The combination of *in vivo* and *in vitro* approaches provided comprehensive insights into parasite-host interactions, shedding light on mechanisms that govern *T. gondii* pathogenesis and transmission.

## Materials and methods

### Ethics statement

Animals were treated in compliance with the guidelines set by the Institutional Animal Care and Use Committee (IACUC) of the University of Wisconsin School of Medicine and Public Health (protocol #M005217), which adheres to the regulations and guidelines set by the National Research Council. The University of Wisconsin is accredited by the International Association for Assessment and Accreditation of Laboratory Animal Care.

### Host cells lines cultivation

All cultures were maintained in a 37°C humidified 5% $CO_2$ incubator unless specified. Briefly, Human Foreskin Fibroblasts (HFF, ATCC SCRC-1041) and Human Colorectal Adenocarcinoma Cells (Caco-2, ATCC HTB-37) were cultured in Dulbecco's modified Eagle's medium (DMEM, Gibco) supplemented with 10% fetal bovine serum (FBS), 2 mM L-glutamine, and 1% penicillin-streptomycin. Human Intestinal Epithelial Cells (HIEC-6, ATCC CRL-3266) were cultured in OptiMEM 1 Reduced Serum Medium (Gibco) supplemented with 4% FBS, 10 mM GlutaMAX (Gibco), 20 mM HEPES (Gibco), and 10 ng/mL Epidermal Growth Factor (EGF). All cultures were periodically tested for mycoplasma and tested negative.

### Experimental mice

Swiss Webster or C57BL/6 female and male mice, 8–12 weeks-old, were used for all the experiments. Infected or moribund mice were euthanized with $CO_2$.

### *Toxoplasma gondii* tachyzoite cultivation

*T. gondii* tachyzoites were cultured in HFF cells and passed every two or three days. To avoid the appearance of any tissue culture specific characteristics, all experimental procedures were done using low passage parasites and used no longer than passage twenty. *T. gondii* strains: a) type I: RH ΔKU80ΔHPT, RH mCherry and RH GFP; b) type II: PruΔHPT:luciferase, ME49 ΔKU80DHPT (donated by Dr. S. Lourido), ME49 mCherry, and M4; and type I/III: EGS LDH2p-GFP/SAG1p-mCherry [40] (donated by Dr. L. Weiss).

### *Toxoplasma gondii in vitro* bradyzoite cultivation

Briefly, HFF cells were infected with $5 \times 10^5 – 2 \times 10^6$ tachyzoites for 2 h. Then, extracellular parasites were removed and differentiation media (RPMI 1640 without sodium bicarbonate (Sigma-Aldrich), 1% FBS, and 40 mM HEPES, at pH 8.1.) was added. Infected cells in differentiation media were incubated at 33°C for up to 12 days in a thermostatic incubator under normoxic conditions with ambient $CO_2$ concentrations. The flask cap was tightly closed to avoid gas exchange. Bradyzoite and cysts formation were confirmed by staining SAG1, BAG1, or DBA. In EGS parasites, bradyzoite differentiation was monitored *in vivo* by the expression of GFP and loss of mCherry by using EVOS FL Auto system (Life Technologies) scope or a live cell incubator (Incucyte, Sartorius). Quantification of *in vitro* encystment was performed by counting at least 100 positive or negative vacuoles. The statistics were performed using one-way ANOVA in the GraphPad Prism software.

### *Toxoplasma gondii in vivo* encystment

To develop chronic infection and collect brain cysts, Swiss Webster or C57BL/6 mice were inoculated intraperitoneally (i.p.) with $1 \times 10^3 – 1 \times 10^4$ of freshly egressed *T. gondii* tachyzoites of M4, PruΔHPT:luciferase, ME49 mCherry, or EGS LDH2p-GFP/

SAG1p-mCherry parasite strains. Mice were euthanized 28 days post-infection, and the brains were removed to feed other mice or to isolate brain cysts.

### Brain cysts isolation

Cysts were purified from infected mouse brains infected for at least 28 days. Brains were harvested and homogenized in ice-cold PBS. Brain material was passed through 18-, 20-, and 22-gauge needles three times to desegregate the brain. To purify cysts, brain material was centrifuged and resuspended in 30% of Percoll-90 (90% Percoll and 10% 1X PBS) in PBS. Cysts were pelleted by centrifugation at $1,000 \times g$ for 15 min, washed with PBS, counted in a hemocytometer.

### Survival curves of *in vitro* bradyzoite-infected mice

Three male Swiss Webster mice per dose were naturally infected with in vitro cysts of EGS strain: doses were $1 \times 10^2$, $1 \times 10^3$, $1 \times 10^4$, or $1 \times 10^5$. For ME49 in vitro cysts, natural infection doses were $1 \times 10^4$ (one mouse) or $7 \times 10^4$ (three mice)". Eight days in differentiation media for EGS and eleven days in differentiation media for ME49, monolayers containing *in vitro* cyst structures were scraped and lysed using a 27-gauge needle. *In vitro* cysts were quantified, adjusted to the corresponding burden, and centrifuged at $500 \times g$ for 10 min. Pellets were resuspended in 100 µl of 2% sucrose and added into 0.5 cm$^3$ fresh white bread cubes. Mice were fasted overnight [70] and consumed the bread containing *in vitro* cysts within a 1-hour window. ME49 mCherry *in vitro* cysts were used as a positive control of chronic infection. For both strains, mice health was monitored daily for health status. At 28 days post-infection, mice were euthanized, and their brains were collected and analyzed for the presence of cysts.

### Villi isolation from infected mice

Swiss Webster mice were subcutaneously infected with oocysts of the M4 strain (type II, generously donated by Jeroen Saeji and David Arranz-Solís). Upon establishment of chronic infection, mice were euthanized, and their brains were isolated to release bradyzoites, following established protocols [55]. Bradyzoites were then propagated in HFF cells through one passage. Subsequently, $1 \times 10^4$ tachyzoites of the M4 strain were infected in mice to develop chronic infection. At least four brains containing cysts were collected and served as food for two healthy mice, following the natural route of infection. Briefly, two mice were fasted overnight [70] and consumed the brains containing *in vitro* cysts within a 1-hour window. Each mouse was fed with two brains, each containing $\sim 2 \times 10^2$ cysts per brain. The number of cysts were determined isolating brains from two infected mice from the same population, as above described. After feeding, mice were euthanized 3 days post-feeding, and their intestines were isolated. Sections of 6–8 cm of jejunum of infected mice was gently washed with ice-cold PBS, opened longitudinally, and mucus layer was removed by scraping superficially with a glass slide as previously described [33]. Villi were then detached by gentle scraping, collected in tubes, and fixed in 4% paraformaldehyde for 30 min. The fixed villi were centrifuged at 100 $\times$ g for 5 min, washed twice with PBS, and processed for immunofluorescence assay. Floating villi were placed in slides and imaged in a confocal microscope (ZeissLSM 800 Laser Scanning Microscope). This experiment was repeated three times to find foci of infection.

### Bradyzoite excystation and pepsin activation

Bradyzoite exteriorization was performed by incubating purified cysts with digestion buffer containing NaCl and pepsin (5 mg/ml pepsin, 100 mg/ml NaCl, 0.14 N HCl, pH 2.1) as described previously [54,56]. Excystation was carried out for 2 min to 2 h. Digestion solution

containing free bradyzoites was neutralized by addition of 1% $Na_2CO_3$ (v/v). Bradyzoites were counted using a hemocytometer, centrifuged at $2,000 \times g$ for 10 min, and resuspended in the corresponding media.

## Mice *in vivo* imaging system

To study *T. gondii* intestinal interactions followed by the natural route of infection, three males and four females Swiss Webster mice were infected by consuming brains containing cysts (refer to figure legends to see the mice number in each time point). Each mouse ingested two brains with a total approximated dose of $4 \times 10^2$-$5 \times 10^2$ brain cysts. The number of brain cysts were determined by isolating brains from two infected mice from the same population, as above described. At 3 or 5 days post-infection, parasitemia in mice was detected by bioluminescence using the *in vivo* imaging system (IVIS, PerkinElmer) as previously reported [37]. Prior to imaging, mice were anesthetized with 4% isoflurane and then injected retroorbital and i.p. with 200 µL of D-Luciferin potassium salt (15.4 mg/mL in PBS). Immediately after the injection, mice were sacrificed, intestines were isolated and exposed to the IVIS detector, and images were collected. Background measurements were determined by injecting uninfected mice with D-Luciferin and collecting the images as previously described. Additional background controls included the imaging of food and pieces of muscle from infected mice. All conditions were repeated at least twice. For the analysis of infected areas, the small intestines was collected the duodenum, jejunum, and ileum which encompass 33 cm of length [71]. Duodenum was considered as the first 2 cm after the pyloric sphincter, jejunum constituted 15 cm after the end of the duodenum, and ileum constituted the last 15 cm before the cecum. Data quantification of the bioluminescent areas was performed by counting the number of events per intestinal region or by drawing a region of interest (ROI) box for each mouse and recording the total number of photons per second (Total Flux) [38].

## Mice intestines immunohistochemistry

Areas of infection in intestines were detected by IVIS. Three to four infected sections of intestines of 5 mm long were cut and immediately fixed in 4% formalin in PBS overnight and dehydrated in 75% ethanol. Paraffin-embedding and sectioning of intestines were performed by request in the Translational Research Initiatives in Pathology (TRIPath lab, University of Wisconsin-Madison). Paraffin-embedded intestines were serial cut in at least 4 sections per slice to facilitate antibody comparisons. Infected sections of the intestine were processed at least three times for detecting parasites in different planes. Giving a total of at least 36 sections of foci of infection per condition. Intestinal sections were incubated at 65°C in a water bath for 1 h, washed with xylene 3 times for 10 min to remove the paraffin, twice for 1 min with 100% ethanol, twice with 95% ethanol in water, once with 75% ethanol in water, and finally in water. The epitopes were retrieved by exposure to the low-pressure setting in a pressure cooker for 6 min in urea buffer (1.25 M Tris Base, 1 M Urea, pH 9.5). Sections were blocked with 3% goat serum in 0.05 M TBS for 1 h. Primary antibody was incubated at 4°C overnight in 0.05 M TBS. Primary antibodies against- SAG1 (donated by John C Boothroyd,1:40, rabbit polyclonal antibody), BAG1 (donated by Louis Weiss, 1:40, rabbit polyclonal antibody), *T. gondii* (Invitrogen, 1:40, rabbit polyclonal antibody). CD31 (BD Pharmingen, 1:40, rat polyclonal antibody), and CD45 (BD Pharmingen, 1:40, rat polyclonal antibody). Dyes included: wheat germ agglutinin Alexa 488 (Invitrogen, 1:100) and DAPI (Millipore Sigma, 1:1000). Secondary antibodies against- rabbit or rat Alexa 594 (Thermo Fisher Scientific, 1:100). Washed 3 times with 0.05 M TBS and mounted in VECTASHIELD antifade mounting medium (VectorLabs). Each antibody was tested in at least twice biological replicates and by duplicate. Wheat germ

agglutinin was preferred over phalloidin to counterstain the embedded tissue due to punctuated and unclear patterns detected in our samples when staining actin. Samples were imaged on a Zeiss Axioplan III (Imager.M2, Carl Zeiss) or a confocal microscope (ZeissLSM 800 Laser Scanning Microscope). 3D projections were reconstructed from z-stack sections using the Zen imaging software (Carl Zeiss).

Quantification of infected areas was conducted by measuring fluorescence intensity. Regions-of-Interest corresponding to the fluorescence signal of SAG1 in the stroma or enterocytes in infected intestines were identified. The integrated pixel intensity of the entire region-of-interest was measured, and background fluorescence was subtracted. Quantitative analysis of images was performed using ImageJ software as described previously [72]. In addition, a quantification of the location of SAG1-positive vacuoles in the stroma or enterocytes was quantified and graphed. The statistics were performed using one-way ANOVA in the GraphPad Prism software.

### *Ex-vivo* infection of villi

Healthy C57BL/6 mice were euthanized, and their intestines were isolated on an ice-cold Petri dish with PBS. Intestines were gently washed with 1× Pen/Strep and 25 µg/ml gentamicin in sterile PBS. Intestines were opened longitudinally, and villi detached as above described. Villi were washed once in antibiotics/PBS solution, centrifuged at $100 \times g$ for 5 min, and resuspended in Leibovitz L-15 media (Thermo Scientific 31415029) supplemented with 10% FBS and 1× GlutaMAX (Gibco). Villi suspension was placed in chamber slides (Thermo Scientific 177372PK) covered by 3 mg/ml rat tail collagen I (Col-I, Corning), 0.5 N sodium hydroxide (Thermo Fisher Scientific), 50 ng/ µl human fibronectin (Millipore), mixed with 7.5 pH 5× PBS, and complete DMEM medium. Villi were challenged with ~$5 \times 10^5$ of RH GFP tachyzoites or EGS LDH2p-GFP/SAG1p-mCherry *in vitro* bradyzoites for 6 h and 24 h. Samples were then fixed with 4% paraformaldehyde and processed for immunostaining. Viability assays were performed at the same time points by using SYTOX Green Nucleic Acid Stain (Thermo Scientific S7020). Samples were imaged on a Zeiss Axioplan III (Imager.M2, Carl Zeiss) or a confocal microscope (ZeissLSM 800 Laser Scanning Microscope). 3D projections were reconstructed from z-stack sections using the Zen imaging software (Carl Zeiss).

### Parasite growth assays

To measure parasite growth, HFF, Caco-2, or HIEC-6 cells were seeded in 12- or 24- well plates. *In vitro*-differentiated or brain cysts-isolated bradyzoites of ME49 mCherry or *in vitro*-differentiated bradyzoites of EGS strain were isolated and excysted as above described using pepsin or by mechanical disruption using 30-gauge needle. Under-confluent cells were infected with $1 \times 10^4 – 1 \times 10^5$ exteriorized bradyzoites. Plates were incubated in the Incucyte incubator for up to 9 days. For comparative assays, all cell types were seeded in the same plate, infected with the same parasites passage, and imaged and quantified under the same conditions as previously described [37,73]. Samples were quantified using the total mCherry area (µm²/image) as previously published [37,73]. The statistics were performed using one-way ANOVA in the GraphPad Prism software.

### Generation of human intestinal microphysiological systems

MPS were fabricated using soft lithography techniques and filled with a collagen/fibronectin matrix to create a molded lumen that was seeded with a monolayer of Caco-2 cells to create an intestine mimic as previously described [35]. Briefly, the LumeNEXT device is created from two stacked PDMS layers, surrounding a removable PDMS rod, with separate inlet and outlet

ports. After the main chamber is filled with extracellular matrix and polymerized, the PDMS rods are removed to form a molded lumen structure that can be seeded with cells. For specific features about MPS fabrication, please refer to Jimenez-Torres and collaborators [74].

Devices were UV-sterilized for 20 min. To promote matrix adhesion to PDMS, chambers were treated with 1% polyethylenimine (MilliporeSigma) in water solution for 10 min, followed by a treatment of 0.1% glutaraldehyde (MilliporeSigma) for 30 min, and then washed 5 times with water. ECM was prepared consisting of 3 mg/ml rat tail collagen I (Col-I, Corning), 0.5 N sodium hydroxide (Thermo Fisher Scientific), 50 μg/μl human fibronectin (Millipore), mixed with 7.5 pH 5× PBS, and complete DMEM medium. The pH of the ECM matrix was adjusted to pH 7.2 before loading the matrix into the central chamber of the device. ECM was polymerized at room temperature for 20 min and then moved to an incubator at 37°C for at least 1 hour. PDMS rods were then removed leaving molded lumen structures within the ECM gel that can be lined with cells.

To seed the lumens with epithelium, Caco-2 cells were resuspended at $20 \times 10^6$ cells/ml of supplemented DMEM. 3 μl of cell suspension was introduced into the lumens through the inlet port. The MPS was incubated upside down for 30 min, then flipped and incubated for 1 h. Nonadherent cells were removed by washing with culture medium, and additional medium was added to the inlet/outlet ports. Devices were incubated for 24–48 hours to permit the formation of a confluent monolayer of cells that mimic human intestinal lumens. Intestinal lumens were infected with $1 \times 10^3 – 5 \times 10^4$ exteriorized bradyzoites. For the quantification of tachyzoite conversion, seven plates were infected with $3 \times 104$ *in vitro* bradyzoites. EGS tachyzoites were switched to the conditions described above and maintained for at least 7 days. Bradyzoite conversion to tachyzoites was monitored in the device using the EVOS FL Auto system and assessed by detecting PVs positive to mCherry/SAG1. Results are shown as a percentage of the average of red vacuoles in each lumen per plate, 36 lumens total were quantified. An additional staining for cell coverture in the lumens was tracked by actin staining using fluorescein phalloidin (Invitrogen). Barrier stability of lumens was tested using 2 mg/ml of 4 kDa FITC-Dextran (Sigma) and monitoring dextran diffusion.

## Immunofluorescence staining

For immunoassays, infected monolayers were processed following modified protocol based on previous works [47,75]. DBA, SAG1, or BAG1 positive vacuoles, infected monolayers were fixed with 4% paraformaldehyde for 30 min or ice-cold methanol for 20 min. Methanol was also used to quench mCherry or GFP fluorescence in parasites. Fixed monolayers were permeabilized with 0.5% Triton X-100 for 5 min and blocked with 3% BSA and 0.2% Triton X-100 for 30 minutes. For stating the cysts structures, we used *Dolichos biflorus* Agglutinin (DBA) which specifically binds the mucin domain of N-acetyl-galactosamine of the CST1 glycoprotein in the cyst wall [76]. DBA was used coupled to rhodamine (Vector laboratories) or fluorescein (Vector laboratories) was diluted in 3% BSA and 0.2% Triton X-100 and incubated for at least 1 h.

For immunostaining infected villi, fixed tissue was processed as previously described [37]. Fixed villi were permeabilized with 0.5% Triton X-100 for 5 min and blocked with 5% BSA for 30 min. Villi were incubated with polyclonal antibodies against- *T. gondii* (Invitrogen, 1:500, rabbit), overnight at 4°C. Secondary antibody Alexa Fluor 488 (Thermo Scientific, 1:1,000, goat anti-rabbit) were incubated for 2 h at room temperature. For GFP or mCherry parasites no antibodies were used for detecting parasites. Villi were counterstained with rhodamine phalloidin (Cytoskeleton, Inc) or fluorescein phalloidin (Invitrogen), DAPI, and mounted using VECTASHIELD. Villi were imaged using a confocal microscope (ZeissLSM 800 Laser Scanning Microscope).

Immunostaining of parasites in MPS followed a modified protocol based on a previous report [35]. Briefly, cells were fixed with 4% paraformaldehyde for 20 min and permeabilized with 0.2% Triton X-100 (Fisher Scientific) for 10 min or ice-cold methanol for 20 min to quench mCherry fluorescence in parasites. Blocking was performed overnight at 4°C in a buffer solution (3% BSA and 0.1% Tween 20 (Thermo Fisher Scientific). Primary antibodies against- SAG1 (1:40, monoclonal mouse antibody DG52) or BAG1 (1:40, rabbit polyclonal antibodies) in buffer solution were added to the infected lumens and incubated at 4°C for 2 days. Secondary antibodies: rabbit Alexa Fluor 488 (Thermo Fisher Scientific, 1:100) and mouse Alexa Fluor 594 (Thermo Fisher Scientific, 1:100) were added to the buffer solution and incubated for 1 day at 4°C. Additionally, viability of cells in lumens were tracked by using CellMask Green Actin (Invitrogen), counterstained with DAPI, and imaged using a Nikon AXR Confocal Microscope.

## Statistical analysis

Data were analyzed (Prism 9.0; GraphPad Software). Statistical significance was assessed using nonparametric Student's t tests when comparing two conditions/groups, and when comparing more than two groups, significance was assessed using one-way analysis of variance (ANOVA).

## Supporting information

**S1 Fig. *T. gondii* localization in Swiss Webster jejunal villi at 3 days post-ingestion of brains with M4 cysts.** All images represent single events captured in independent infected mice. (A) Schematic representation of (i) the villus and its structures and (ii) the experimental setup; and iii) a confocal image of an uninfected jejunal. (B-G) Confocal images of infected villus with *T. gondii* after 3 days post-ingestion of cysts-containing brains. *T. gondii* is localized in the intestinal lumen (B), inside an enterocyte (C), between cell-to-cell junctions (D), in the jejunal stroma (E, F), and within a potential immune cell (G). In all images (A-G), villi are stained for actin (red, rhodamine phalloidin), *T. gondii* (green, polyclonal antibody), and nuclei (blue, DAPI). Schematic representation in (A) was created with BioRender.com.
(PDF)

**S2 Fig. Distribution of *T. gondii* infection by IVIS in the small intestine of mice fed brains containing cysts.** (A) Schematic representation of mice intestine that was considered for the study. (B) IVIS images of one male and one female mouse fed brains containing cysts at 5 days post-ingestion, and one female after 3 days post-ingestion. Mice were fed with sunflower seeds to avoid an unexpected background. B' inset in the top figure shows a bumped area of the intestine that corresponds with the bioluminescent region detected by IVIS. (C) An uninfected mouse and female mice fed brains containing cysts and small intestine after 3- or 5-days post-ingestion. (D) An uninfected mouse and male mice fed brains containing cysts and small intestine after 3- or 5-days post-ingestion. Schematic representation in (A) was created with BioRender.com.
(PDF)

**S3 Fig. Detection of tachyzoite and bradyzoite markers in the jejunum of mice fed brain containing cysts at 3- and 5-days post-feeding.** (A) Representative epifluorescent images of jejunum at 3 days post-infection showing SAG1-positive and BAG-negative parasites in the stroma. (B) Representative epifluorescent images of jejunum at 5 days post-infection showing SAG1-positive and BAG-negative parasites in the stroma. Intestines are stained for WGA (green), SAG1 (red), BAG1 (red), and nuclei (blue, DAPI).
(PDF)

**S4 Fig. *T. gondii* infection in the jejunum of mice fed brain containing cysts at 3 days post-feeding.** (A) Representative epifluorescent images of jejunum showing SAG1-positive parasites in the stroma and close to the muscular layer. Bottom insets in (A) show a possible immune cell infected by *T. gondii*. (B) Representative confocal images of jejunum showing parasites replicating in the stroma. Parasites were stained using polyclonal antibodies against *T. gondii*. (C) Representative confocal 3D projection of jejunum showing SAG-1 positive parasites replicating in the stroma. Intestines are stained for WGA (green), SAG1 (red), polyclonal antibodies against *T. gondii* (green), and nuclei (blue, DAPI).
(PDF)

**S5 Fig. *T. gondii* infection in the jejunum of mice fed brain containing cysts at 5 days post-feeding.** (A) Confocal images of female and male jejunum showing *T. gondii* parasites replicating near areas with excessive cell infiltration. (B) Confocal image of tachyzoites replicating surrounded by CD45-positive cells. (C) 3D projection of tachyzoites replicating within the stroma. (D) Epifluorescent image of tachyzoites (red) near the endothelium. RBC states for red blood cells. (E) Representative epifluorescent images of uninfected jejunum stained against CD45. (F) and (G) Show CD31-positive cells, as a result of our best attempt to obtain CD31 in the jejunal villi after 3 days post-feeding. Intestines are stained for polyclonal antibodies against *T. gondii* (green), CD31 (red), CD45 (green), and nuclei (blue, DAPI).
(PDF)

**S6 Fig. *In vitro* bradyzoite differentiation of *T. gondii* at 33°C.** (A) Fluorescent imaging of HFF cells infected with ME49 mCherry for 8-days post-infection in HFF media or differentiation media. (B) Fluorescent images of DBA-, BAG1-, SAG1-positive parasites/cysts after 6- or 12-days post-switching to differentiation media. (C) Fluorescent images of DBA- or BAG1-positive Pru parasites/cysts after 12-days post-switching to differentiation media. (D) Fluorescent images of DBA-, BAG1-, SAG1-positive RH parasites/cysts after 2-, 7- and 12-days post-switching to differentiation media. (E) Fluorescent images of LDH-2p/GFP, SAG1/mCherry or DBA-positive parasites/cysts after 7- or 12-days post-switching to differentiation media. (F) Fluorescent images of Caco-2 and HFF infected with in vitro-differentiated bradyzoites at 72 hours post-infection. (G) Representative image of a brain cyst isolated from an infected mice fed with in vitro-generated bradyzoites after 28 days post-feeding.
(PDF)

**S7 Fig. In vitro differentiation of EGS strain at 33°C.** Alive fluorescent imaging of EGS parasites detecting mCherry-expressing tachyzoites or GFP-expressing bradyzoites. (A) Representative fluorescent images of HFF cells infected with EGS parasites at 3 days in differentiation media at 37°C, showing mCherry-expressing tachyzoites. (B) Representative fluorescent images of HFF cells tracking the differentiation of EGS parasites into GFP-expressing bradyzoites under our differentiation protocol at 33°C from one to nine days post-infection.
(PDF)

**S8 Fig. Viable *in vitro*-differentiated bradyzoites do not infect Caco-2 in MPS.** (A) Fluorescent images showing no replication of *in vitro* differentiated ME49 mCherry bradyzoites in the Caco-2 lumen after 5- or 6- days post-infection. (B) Fluorescent images of ME49 mCherry parasites replicating in Caco-2 after 3 days post-infection, as a positive control of parasite viability. They infected as *in vitro* differentiated bradyzoites in MPS. (C) Determination of cell viability of *in vitro* differentiated bradyzoites after pepsin digestion by using SYTOX-green. (D) Caco-2 lumens infected with ME49 mCherry tachyzoites used as a control of infection. Punctuated lines represent the edges of the lumen and its boundary with the matrix.
(PDF)

**S9 Fig.** *In vitro*-**differentiated EGS bradyzoites are inactivated and they do not infect Caco-2 in MPS.** Fluorescent images showing no replication of EGS *in vitro* differentiated bradyzoites in the Caco-2 lumen after up to 5- days post-infection.
(PDF)

**S10 Fig. Bradyzoite replication and conversion to tachyzoites in 2D cell culture.** (A) Representative fluorescent images of parasite replication in HFF, Caco-2, and HIEC-6 cells infected with ME49 mCherry brain bradyzoites, either activated by pepsin for 30 min or subjected to mechanical lysis, at 7 days post-infection. Graph shows parasite kinetics of growth in HFF, Caco-2 and HIEC-6 cells for pepsin-activated versus inactivated bradyzoites. (B) Representative fluorescent images of tachyzoite conversion in HFF infected with EGS *in vitro* differentiated bradyzoites activated by pepsin for 30 min. Conversion and replication were followed up to 4 days post-infection. Top graph shows parasite kinetics of growth in the HFFs cells activated by pepsin for 2 or 30 minutes or inactivated. Bottom graph shows EGS bradyzoite conversion to tachyzoites kinetics in the HFFs cells activated by pepsin for 30 minutes.
(PDF)

**S11 Fig. Methanol quenched mCherry fluorescence in human microphysiological devices.** (A) Show a lumen infected with ME49 mCherry brain bradyzoites after 48 hours post-infection. Top lumen shows unquenched parasites. Bottom lumen shows the same lumen with the mCherry quenched after incubation with methanol for 20 min. Bottom lumen was over exposed to detect any remnant of mCherry in the lumen. (B) Show a lumen infected with ME49 mCherry brain bradyzoites after 72 hours post-infection. Top lumen shows unquenched parasites. Middle lumen shows the same lumen with the mCherry quenched after incubation with methanol for 20 min. Bottom lumen shows an IFA against SAG1 and BAG1. Middle and bottom lumen were over exposed to detect any remnant of mCherry or GFP in the lumen.
(PDF)

**S12 Fig. Explants of jejunal villi cultured for 24 hours post-isolation.** (A) Intact villus surrounded by villi material. (B) Free floating enterocytes due to the disintegration of villi during incubation. (C) Cellular viability assessed using SYTOX Green, with dead cells highlighted by green-stained nuclei.
(PDF)

**S1 Movie. Tachyzoites replicating in the stroma of jejunal villi of mice fed cysts containing cysts after 3 days post-ingestion.**
(MP4)

**S2 Movie. Tachyzoites localizing close the area full of infiltrated after 5 days post-ingestion.**
(MP4)

**S3 Movie. ME49 mCherry brain bradyzoites activated by pepsin for 30 min and infecting HFFs cells for up to 6 days.**
(MP4)

**S4 Movie. ME49 mCherry brain bradyzoites activated by pepsin for 30 min and infecting Caco-2 cells for up to 6 days.**
(MP4)

**S5 Movie. ME49 mCherry brain bradyzoites activated by pepsin for 30 min and infecting HIEC-6 cells for up to 6 days.**
(MP4)

**S6 Movie. HIEC-6 migrating in collagen I/fibronectin in the MPS.**
(MP4)

## Acknowledgments

We thank Louis Weiss for the EGS strain, Jeroen Saeji and David Arranz-Solís for the M4 oocysts, and Sebastian Lourido for ME49ΔKU80ΔHPT strain. We thank members of the Knoll laboratory for their helpful discussions. We thank Michael Panas and John Boothroyd for providing the DG52 hybridoma for anti-SAG1 monoclonal antibody production. We thank the UW Small Animal Imaging and Radiotherapy Facility (SAIRF) for providing access to the IVIS imaging system, we thank the UW Optical Imaging Core (UWOIC) for use of the Nikon AXR confocal microscope (NIH grant number 1S10O34394-01), and we thank the Translational Research Initiatives in Pathology (TRIPath lab) for processing our intestinal samples. We thank members of the Beebe lab for their technical help and helpful discussions. We specially thank Imran Khan for his technical support.

## Author contributions

**Conceptualization:** Carlos J. Ramírez-Flores, David J. Beebe, Laura J. Knoll.

**Funding acquisition:** Nicole D. Hryckowian, David J. Beebe, Sheena C. Kerr, Laura J. Knoll.

**Investigation:** Carlos J. Ramírez-Flores, Nicole D. Hryckowian, Andrew N. Gale, Kehinde Adebayo Babatunde, Marcos Lares, Laura J. Knoll.

**Methodology:** Carlos J. Ramírez-Flores, Nicole D. Hryckowian, Laura J. Knoll.

**Project administration:** Laura J. Knoll.

**Supervision:** Laura J. Knoll.

**Validation:** Carlos J. Ramírez-Flores, Andrew N. Gale.

**Visualization:** Carlos J. Ramírez-Flores, Nicole D. Hryckowian.

**Writing – original draft:** Carlos J. Ramírez-Flores.

**Writing – review & editing:** Carlos J. Ramírez-Flores, Nicole D. Hryckowian, Andrew N. Gale, Kehinde Adebayo Babatunde, David J. Beebe, Sheena C. Kerr, Laura J. Knoll.

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
