## [Decision Letter · Decision Letter 0]

28 Oct 2024

PNTD-D-24-01427Host-parasite intestinal interactions: Toxoplasma cyst conversion and intestinal infection in mice and human microphysiological systemsPLOS Neglected Tropical Diseases Dear Dr. Knoll, Thank you for submitting your manuscript to PLOS Neglected Tropical Diseases. After careful consideration, we feel that it has merit but does not fully meet PLOS Neglected Tropical Diseases's publication criteria as it currently stands. Therefore, we invite you to submit a revised version of the manuscript that addresses the points raised during the review process. Please submit your revised manuscript within 60 days Dec 27 2024 11:59PM. If you will need more time than this to complete your revisions, please reply to this message or contact the journal office at plosntds@plos.org. Please include the following items when submitting your revised manuscript:* A rebuttal letter that responds to each point raised by the editor and reviewer(s). You should upload this letter as a separate file labeled 'Response to Reviewers '. This file does not need to include responses to any formatting updates and technical items listed in the 'Journal Requirements' section below.* A marked-up copy of your manuscript that highlights changes made to the original version. You should upload this as a separate file labeled 'Revised Manuscript with Track Changes '.* An unmarked version of your revised paper without tracked changes. You should upload this as a separate file labeled 'Manuscript '. If you would like to make changes to your financial disclosure, competing interests statement, or data availability statement, please make these updates within the submission form at the time of resubmission. Guidelines for resubmitting your figure files are available below the reviewer comments at the end of this letter. We look forward to receiving your revised manuscript. Kind regards, Sarah EwaldGuest EditorPLOS Neglected Tropical Diseases Laura-Isobel McCallSection EditorPLOS Neglected Tropical Diseases

Shaden Kamhawi

co-Editor-in-Chief

Paul Brindley

co-Editor-in-Chief

 **Journal Requirements:** **Additional Editor Comments (if provided):** Overall, this is a significant body of work that provides new methodology to study host-parasite interactions that will be broadly useful to the field. There was high excitement for the study despite some major concerns about the depth and rigor of the analyses. It is possible that most of the reviewers concerns can be addressed by modifying some of the conclusions, re-organizing some of the data in the manuscript and by performing more in depth analyses of existing data. However, since many figures lack information about the number of biological replicates and statistical analysis, additional experimentation may be necessary.

Please address issues brought up by reviewer 1 regarding missing information in the figure legends and materials and methods that are necessary to interpret the data shown in Figures 1-3 and the associated supplemental data. There are several places where more rigorous analyses of existing data may need to be added to support conclusions drawn from the data or ensure that the data is correctly analyzed. Quantification should be based on individual mice where a similar number of observations (eg. fields of view or luc+ nodes) are averaged for each mouse as a single point and stat are performed on mean of mice in 1 experimental group compared to another as SEM not SD (eg Fig 2E, Fig 3D, ).

Throughout the paper the N (including # of biological replicates, # of events/objects/fields of view quantified) and statistical analysis (for graphs) should be reported for each Figure panel.

Both reviewer 1 and reviewer 2 were concerned about conclusions drawn from comparing gavage with cyst feeding oral infection techniques which should be addressed by altering language or providing data using the same inoculation approach. Unless the infection route is controlled for, please avoid making direct comparisons between data derived from distinct infection protocols.

The paper also may benefit, as suggested by reviewer 1, by moving data that are not sufficiently powered for statistical analysis to supplemental data and summarizing images with statistical analysis wherever possible. This includes but is not limited to:

-Figure 4 Pru and RH

-Fig 5 SAG1 vs. LDH2 staining frequencies analysis

-Fig6 A-B summary analysis of representative images, C-D stats of surgical curves N and number of unique experimetns49mCherry replication/cyst frequency

-Fig 7B-C stats and # of experiments

-Fig 8-10 quantification with stats and # of experiments.**Reviewers' Comments:** Reviewer's Responses to Questions

**Key Review Criteria Required for Acceptance?**

**Methods**

-Are the objectives of the study clearly articulated with a clear testable hypothesis stated?

-Is the study design appropriate to address the stated objectives?

-Is the population clearly described and appropriate for the hypothesis being tested?

-Is the sample size sufficient to ensure adequate power to address the hypothesis being tested?

-Were correct statistical analysis used to support conclusions?

-Are there concerns about ethical or regulatory requirements being met?

Reviewer #1: In general the experimental methods are quite clear, however there are many occurrences where method specific details are lacking. For example number of samples, number of mice, ratio of positive to negative samples, appropriate controls.

How the mice were infected with a consistent number of cysts to ensure robust, consistent infections that can be compared between samples and sexes when the cysts are brain derived is missing from the manuscript and is vital information to draw conclusions comparing strains, sexes and distributions. This information is clear for in vitro derived cyst infections.

Reviewer #2: This is a very useful paper that provides interesting information and new data about bradyzoites, differentiation in vitro and the mechanism of bradyzoite transport across epithelial borders in the intestine. The statistics and sample sizes are adequate for the data presented. All regulatory requirements have been met.

A minor correction in the introduction line 67 is that Toxovax is a licensed attenuated strain (S48 strain) veterinary vaccine for sheep for T. gondii.

**Results**

-Does the analysis presented match the analysis plan?

-Are the results clearly and completely presented?

-Are the figures (Tables, Images) of sufficient quality for clarity?

Reviewer #1: I have broken my review into what I consider the three big results sections of the manuscript:

1. in vivo interactions in the gut:

The first 6 figures of this manuscript do not fit well with the title and appear to be included to justify the use of MPS to better probe cyst gut interactions. I would suggest that these be included as a single main text figure explaining the challenges of in vivo systems with supporting figures in the supplementary data if necessary. Including it ‘as is’ detracts from the exciting data presented in the later figures using MPS, that more closely fit the paper title.

If the data are to remain in their current format the text needs to be clarified to state; the numbers of observations and samples analyzed, and the following specific issues need to be addressed:

The data presented in figures 1 – 3 of this manuscript provide new observational information about parasite infection of the gut in the first 3 – 5 days post infection. It should be clarified that this is a limited number of observations that cannot be quantified, and the number of animals, individual samples (e.g. number of villi analyzed, number of foci imaged etc.) and specific methodologies need to be included.

For Figure 1. I cannot discern how many mice had their villi scraped, how many villi were analyzed to detect parasites and I’m unclear if the 6 examples shown are the only parasites found or if these are representative images of parasites in those specific cells and quantification could be included. Are heavily infected villi missing from the analysis because they are more fragile and therefore lost? Or do the authors believe this low density of infection is the natural occurrence? How many cysts were used to establish the infection? Could optimizing the number of cysts or the preparation of cysts increase the number of parasites per villi or the number of usable villi that can be obtained.

For Figure 2 and supplementary Figure 1. IVIS images from 1 uninfected and 2 male and 2 female mice are shown for each time point and more intestines that have been isolated. However, the numbers that these are representatives for is not included. Likewise for the number of mice for quantifications in D and E, for each of the 2 experiments how many male and female mice were analyzed, and were all foci identified used for flux quantifications? To draw broad conclusions about distribution throughout the gut and the number of foci, several mice (5+ of each sex, in my opinion) would have been analyzed and the infection dose would have to be consistent, or the differences could be due to infection dose rather than mouse sex as implied in the text. More detail is required for these experiments.

Figure 3 and Supplementary figures 2 – 4. Like with figure 1 this is interesting data that shows tachyzoites are the only stage present in the gut at 3- and 5-days post infection and that these parasites can be found in different locations and might cause changes to the gut (S4). Also, like Figure 1 quantification of the frequency of these observations is missing. Additionally, how the infections were controlled to be comparable between male and female mice, between time points and replicates is not included in the legend or methods.

The bumps associated with foci of parasite replication are also interesting, but quantification of how often a bump is found and the number of slices that were analyzed to determine they are not found in uninfected mice and that they are only found at “replication/infection hotspots” is needed.

The low number of observations in these experiments is a clear justification for why the MPS is such a good way to look at the initial interactions but it is my opinion that the narrative would benefit from condensing and clarifying the message of these data.

2. Optimization of an in vitro differentiation assay.

The optimization of the in vitro differentiation assay to yield such high differentiation rates is going to be really useful to the field and the data seem robust. As with the previous set of comments I also believe there are too many examples and not enough quantification. And would suggest including one representative image of how the quantification was done is sufficient for each staining method (BAG1 or DBA) or fluorescent line (EGS) accompanying the quantification making a single figure. The images from different time points and different lines with the same methodology are overwhelming and doesn’t add much.

(minor) Could it be made clear for each condition not only the temperature and medium the parasites are grown in but also the CO2 level.

(minor) Line 258/9 what does this statement mean?

Figure 6. A) ME49 mCherry constitutivey expressed mCherry, how do you discern they are bradyzoites replicating in the HFF and Caco-2 cells?

(Exp) C/D If I am reading the legend correctly and only 3 (feeding, EGS) or 2 (gavage, ME49) mice were infected with each dose for each strain. This is not enough mice to draw conclusions about virulence. The two infection methods also cannot be compared as previous work has shown infection route changes dissemination and virulence. Furthermore, there is not enough context for these infections, how many in vivo derived cysts are required for mice to succumb to infection with each of these lines? To make the statement (figure title) that “in vitro generated bradyzoites exhibit virulence similar to that of brain cysts” this is required. The text also suggests ME49 was also fed to mice and a mouse (out of an unknown number) did become chronically infected with a dose of 1x104 cysts.

Supplementary Figure 6. This section of the text is confusing and I’m not sure what the goal was or the conclusion. The opening sentence seems quite misleading.

3. Modeling parasite-gut interactions with MPS.

To me these are the most important and exciting data presented in the manuscript and should be the major focus based on the title. I think the formatting of these sections is much clearer than the previous ones and the only overall request is further details on how experiments were standardized to allow comparisons. More specific comments:

The MPS acronym is used widely without explanation of what it stands for/is outside of the title, which does not include the (MPS). I would like a little expansion of the system lines 288-295.

298. How do you ensure the me49 mCherry line is a pure bradyzoite line, as far as I am aware, mCherry expression is constitutive in this line and it is only the EDS that has stage specific expression of fluorescent proteins.

Figure 7B quantification of mCherry area after seeding MPS with acid pepsin digested bradyzoites shows low fluorescence at days 4—6 and really the signal isn’t robustly detected of different until day 8 post infection, Supplementary 7 shows images at 5 and 6 dpi, based on the quantification in the main figure I would not have expected to see parasite growth, so the inclusion seems odd. The label I believe should read dpi and says 5hpi.

(Exp) Figure 8A these experiments use ME49 mCherry parasites then also use SAG1 with alexa 594 to “stage” parasites. If mCherry signal is completely lost with the authors fixing method, then this could be acceptable, but no controls are provided to show this. The resolution is not high enough to determine if the signal is peripheral (a would be expected for SAG1) or cytosolic (as would be expected for mCherry). The co-labelling could just be BAG1 and residual mCherry, the SAG1 with Alexa 594 secondary needs more controls to be robust.

8B The resolution of the images shown is difficult to be sure that there is no remaining GFP signal from bradyzoite, especially with the auto fluorescence seen in the MPS. The dynamics presented in this figure are interesting and the technology provides a unique way to look at this that has been inaccessible in vivo. However, there is no quantification and the kinetics and dynamics seen here in one CACO-2 MPS needs to be repeated and quantified to present a stronger finding.

Figure 9 reads like an extension of Figure 1 and doesn’t flow from the previous MPS results. It is also observational, like Figure 1.

Figure 10. This is a very elegant way to look at this transmigration and infection of an adjacent MPS. For 10B was the seeding density very different to Figure 8 as there seem to be vastly different numbers of parasites and few to no bradyzoites on day 3 post infection in figure 8, whereas both stages are clearly visible in Figure 10, a lot of the red signal for tachyzoites expressing mCherry also looks artefactual, better representative images might be needed. There are structures in 10B zoom in that don’t appear in the lower resolution image (the red channel mostly, right hand side).

10C the tachyzoites migrating through the matrix and to the other uninfected lumen is fantastic, it does appear that there are parasites visible by brightfield that don’t show up as mCherry positive, is this an imaging issue (not bright enough) or are these still bradyzoites or intermediates? It is convincing that these parasites migrate through the matrix and into the uninfected lumen. It looks like this occurs when an enormous number of parasites are found in one region (based on this one picture) is this a prerequisite or do parasites migrate into the matrix even if the density within the CACO2 cells it low?

Reviewer #2: The IVIS experiments are very interesting and, in general, support previous observations suggesting infection is more common in the jejunal region of the intestine.

The migration experiments are clearly presented and will be of interest to the scientific community as will the cell culture system used (MPS)

The observation of the value of low temperature 33C on stage conversion is important as this may provide an improved way of obtaining in vitro cysts for both infection and other studies on this life cycle stage. Do you have any data on the gene expression in these low temperature cysts compared to those generated at 37C (e.g. RNAseq or scRNAseq)? Providing such date would be useful and interesting with regards to any differences that exist. Another alternative would be to look for RNA expression of genes only seen in cysts in vivo such as BCP1 (PMC5161674).

Specific Comments

Line 191. Please expand why actin staining was not useful (usually actin epitopes can be seen in paraffin material treated in the manor described in your methods section).

Line 252. I doubt the GFP cysts are expelled from host cells, but more likely host cells detach from the substrate and then are floating. A way to address this would be if the cysts have both a host cell membrane and cyst wall (TEM of the free floating cysts would demonstrate that 2 membranes, host cell plasma membrane and parasitophorous vacuole membrane surround the cyst.

Line 275 to 285. IVIS in an intact animal really needs red shifted fluorescence, to that end it is not surprising that GFP was not detectable from the normal tissue signal and mCherry is not an ideal red shifted fluorochrome for this technique either in intact mice. Furthermore, the use of Alfalfa-free mouse chow is needed to decrease background fluorescence (is this what is referred to on line 283 as specialized diet) and is usually done for 7 day to make sure the gut is free anything that would cause autofluorescence.

Line 482-483 Why would the use of gut microorganisms be useful? Is there any data on a role for this in other pathogens and differentiation?

Line 582 to 595 I agree that direct oral gavage makes sense, in that putting cysts into the mouth on bread might allow infection in the oral cavity without the cysts getting into the intestine. To that end, why is the positive control by gavage and the experimental condition by feeding on bread? Should not the exact same route been used for both the control and experimental group?

Line 653-655: Please provide the source for the various antibodies used (SAG1 rabbit, BAG1 rabbit, T. gondii polyclonal rabbit, CD31 rat and CD45 rat.

**Conclusions**

-Are the conclusions supported by the data presented?

-Are the limitations of analysis clearly described?

-Do the authors discuss how these data can be helpful to advance our understanding of the topic under study?

-Is public health relevance addressed?

Reviewer #1: With the technical limitations, low numbers of sample and minimal quantification I find the statements in the discussion to be overreaching, especially when talking about the data from in vivo samples. Pepsin activation did not appear essential for in vitro culture in CACO2 cells in monolayer format, only in MPS. This is not adequately discussed.

The Discussion however is quite reflective of the observations using the MPS for stage transition kinetics and migration.

Reviewer #2: The data are well organized and support the discussion and analysis.

**Editorial and Data Presentation Modifications?**

Reviewer #1: The manuscript needs a thorough editing for which data should be included in the main body and what should be in the supplements. At present the key messages of the work are lost because all data relating to the project seem to be included in the main document with duplications in the supplement.

In addition there are grammatical and language issues throughout and a specialist edit for this after reformatting may be needed.

Reviewer #2: None

**Summary and General Comments**

Reviewer #1: There is some exciting data presented in this work using novel technologies to give insight into the earliest interactions between T. gondii and the gut. The manuscript, however, lacks a clear narrative and suffers from over inclusion of optimization data, duplications of observational data and a lack of quantification and experimental details making the broader interpretation of the data difficult, especially in the context of the title and goals of the work.

I believe that the underlying data is of good quality, but a thorough reformatting of the manuscript is required. Frequently the supplementary data doesn’t show anything different to the figure in the main text, just provides an additional example of the same data which I don’t think is necessary and makes the narrative harder to follow.

I have tried to outline the areas where the manuscript narrative could be altered for clarity. The two required experiments are also highlighted with (Exp). All others, I believe, can be addressed by providing additional details and within the writing of the manuscript.

Reviewer #2: A very interesting paper with lots of data that will be of interest to people studying Toxoplasma differentiation and the early stages of infection via the oral route of infection.

PLOS authors have the option to publish the peer review history of their article (what does this mean? ). If published, this will include your full peer review and any attached files.

**Do you want your identity to be public for this peer review?** For information about this choice, including consent withdrawal, please see our Privacy Policy .

Reviewer #1: No

Reviewer #2: No

---

## [Decision Letter · Decision Letter 1]

20 Jan 2025

Dear Prof. Knoll,

We are pleased to inform you that your manuscript 'Modeling Toxoplasma gondii-gut early interactions using a human microphysiological system' has been provisionally accepted for publication in PLOS Neglected Tropical Diseases.

Best regards,

Sarah Ewald

Guest Editor

Laura-Isobel McCall

Section Editor

Shaden Kamhawi

co-Editor-in-Chief

Paul Brindley

co-Editor-in-Chief

I am delighted to inform you that we are happy to accept the revised manuscript for publication. There are a few, minor edits suggested by Reviewer 1 that we kindly request you address during editorial revision. These re included below. Congratulations on completion of this body of work which is bringing valuable observations and new research tools to the community.

Reviewer's Responses to Questions

**Key Review Criteria Required for Acceptance?**

**Methods**

-Are the objectives of the study clearly articulated with a clear testable hypothesis stated?

-Is the study design appropriate to address the stated objectives?

-Is the population clearly described and appropriate for the hypothesis being tested?

-Is the sample size sufficient to ensure adequate power to address the hypothesis being tested?

-Were correct statistical analysis used to support conclusions?

-Are there concerns about ethical or regulatory requirements being met?

Reviewer #1: (No Response)

Reviewer #2: My concerns have been addressed in this revision. The paper is clear and the data are interesting and clearly discussed and presented. The methods section provides sufficient information to replicate the study.

**Results**

-Does the analysis presented match the analysis plan?

-Are the results clearly and completely presented?

-Are the figures (Tables, Images) of sufficient quality for clarity?

Reviewer #1: (No Response)

Reviewer #2: Yes, the data, results and figures are clear.

**Conclusions**

-Are the conclusions supported by the data presented?

-Are the limitations of analysis clearly described?

-Do the authors discuss how these data can be helpful to advance our understanding of the topic under study?

-Is public health relevance addressed?

Reviewer #1: (No Response)

Reviewer #2: The conclusions are supported by the paper

**Editorial and Data Presentation Modifications?**

Reviewer #1: line 154 - I think this should be changed to euthanized to be in line with the rest of the manuscript and accepted terminology.

190-191 while I agree this could be due to lack of replication it could also simply be below the limit of detection for this method.

289-291 The Authors have done a tremendous job of adding in all of the animal numbers and numbers of examples analyzed for each of the experiments the mCherry infections where 1 and 3 mice showed infection is the only example where the total number challenged is missing - I have assumed that it is the same as EGS and therefore 3 mice but it would be great if it could be added for completeness.

319 - Type, Figure S8B/S6B

346 - Typo Figure 3CSAG

Figure 4B, lines 362-369 Took me a little while to interpret this figure, I think it is cumulative % SAG1 positive based on the text but a little more detail in the legend would be helpful to aid interpretation of this, for ease it could also be made 4C.

Reviewer #2: No further modifications are suggested

**Summary and General Comments**

Reviewer #1: The reviews have done an amazing job of reformatting the manuscript and figures and I thank them for the time and effort they put in to address my comments. The work presented is now very clear and the findings come across as more impressive for it.

Reviewer #2: Overall, a very nice paper that adds to the literature and contains important data on both an intestinal model of T. gondii as well as an improved method to obtain tissue cysts from in vitro cultures.

PLOS authors have the option to publish the peer review history of their article (what does this mean? ). If published, this will include your full peer review and any attached files.

**Do you want your identity to be public for this peer review?** For information about this choice, including consent withdrawal, please see our Privacy Policy .

Reviewer #1: No

Reviewer #2: No

---

## [Editor Report · Acceptance letter]

Dear Prof. Knoll,

We are delighted to inform you that your manuscript, "Modeling Toxoplasma gondii-gut early interactions using a human microphysiological system," has been formally accepted for publication in PLOS Neglected Tropical Diseases.

Best regards,

Shaden Kamhawi

co-Editor-in-Chief

Paul Brindley

co-Editor-in-Chief
